

# Evapotranspiration partition using the multiple energy balance version of the ISBA-A-g$_s$ land surface model over two irrigated crops in a semi-arid Mediterranean region (Marrakech, Morocco)

Aouade Ghizlane[1], Jarlan Lionel[2,*], Ezzahar Jamal[3,4], Er-raki Salah[5,4], Napoly Adrien[6], Benkaddour Abdelfattah[1], Khabba Said[7,4], Boulet Gilles[2], Garrigues Sébastien[8,9], Chehbouni Abdelghani[2,4], Boone Aaron[6]

[1]Laboratoire des Géo- ressources/LMI TREMA, Faculté des Sciences et Techniques, Université Cadi Ayyad, Marrakech, Maroc.
[2]Centre d'Etudes Spatiales de la Biosphère (CESBIO)/IRD, Toulouse, France.
[3]Equipe de Mathématiques et traitement de l'information (MTI), Ecole Nationale des Sciences Appliquées, Université Cadi Ayyad, Safi, Maroc.
[4]CRSA, Center of Remote Sensing Application, Mohammed VI Polytechnic University UM6P, Benguerir, Morocco.
[5]LP2M2E, Faculté des Sciences et Techniques, Université Cadi Ayyad, Marrakech.
[6]Centre National de Recherches Météorologiques (CNRM), Météo-France/CNRS, Toulouse, France.
[7]LMME, Département de physique, Faculté des Sciences Semlalia, Université Cadi Ayyad, Marrakech, Maroc
[8]EMMAH, INRA, Université d'Avignon et des Pays de Vaucluse, Avignon, France
[9]Centre for Ecology and Hydrology (CEH) Wallingford, UK

*Correspondence to*: Lionel Jarlan (lionel.jarlan@cesbio.cnes.fr)

**Abstract.** The main objective of this work is to question the representation of the energy budget in surface-vegetation-atmosphere transfer (SVAT) models for the prediction of the convective fluxes in the case of irrigated crops with a complex structure (row) and under strong transient hydric regimes due to irrigation. To this objective, the Interaction Soil-Biosphere-Atmosphere (ISBA-A-gs) based on a composite energy budget (named hereafter ISBA-1P for 1 patch) is compared to the new multiple energy balance (MEB) version of ISBA using two representations of the canopy energy budget: a coupled approach (ISBA-MEB) where the vegetation layer is located above the soil and a patch representation corresponding to two-adjacent uncoupled source schemes (ISBA-2P for 2 patches). The evaluation is performed over a winter wheat field, taken as an example of homogeneous canopy and on a more complex open olive orchard. Continuous observations of evapotranspiration (ET) with Eddy covariance system, soil evaporation (E) and plant transpiration (T$_r$) with Sapflow and isotopic methods were used to evaluate the three representations. A preliminary sensitivity analyses showed a strong sensitivity to the parameters related to turbulence in the canopy introduced in the new ISBA-MEB version. The ability of the single and dual-source configuration to reproduce the composite soil-vegetation heat fluxes was very similar: the RMSE differences between ISBA-1P, -2P and -MEB did not exceed 10 W/m$^2$ for the latent heat flux. These results showed that a composite energy balance on homogeneous covers is sufficient to reproduce the total convective fluxes. By contrast, differences were highlighted on the partition of ET. In particular, the ISBA-2P version showed an over-estimation of soil





evaporation of about 20% because of a direct exposition to incoming solar radiation and because there is no root extraction for the bare soil patch with regards to –MEB and -1P representations. By contrast, the dual source configurations including both the uncoupled (ISBA-2P) and the coupled (ISBA-MEB) representations outperformed the single source version (ISBA-1P) with slightly better results for ISBA-MEB in predicting both total heat fluxes and evapotranspiration partition over the

moderately open canopy of the Olive orchard site. Concerning plant transpiration in particular, the coupled approach ISBA-MEB provides better results than ISBA-1P and, to a lesser extent ISBA-2P with RMSEs of 1.60, 0.90, 0.70 mm/day and R² of 0.43, 0.69 and 0.70 for ISBA-1P, -2P and MEB respectively. In addition, it is shown that the acceptable predictions of composite convective fluxes by ISBA-2P for the Olive orchard are obtained for the wrong reasons as neither of the two patches is in agreement with the observations because of a bad spatial distribution of the roots and of a lack of incoming

radiation screening for the bare soil patch. This work shows that composite convection fluxes predicted by the SURFEX platform as well as partition of evapotranspiration in a highly transient regime due to irrigation is improved for moderately open tree canopies by the new coupled dual-source ISBA-MEB model. It also points out the need for further local scale evaluation on different crops of various geometry (more open rainfed or denser intensive olive orchard) to provide adequate parameterization to global data base such as ECOCLIMAP-II in the view of a global application of the ISBA-MEB model.


Keywords: ISBA model, Evapotranspiration, Crop transpiration, Soil evaporation, Eddy covariance, Sapflow, Stable isotopes, Flood-irrigated crops, Semi-arid region.

**1 Introduction**

As the only connection linking the water budgets and energy balance, the evapotranspiration (ET) is a primary process

driving the moisture and heat transfers between the land and the atmosphere (Xu et al., 2005; Xu and Singh, 2005; Wang et al., 2013). A good prediction of ET is thus of crucial importance for water recycling processes (Eltahir, 1996) and, in fine, for numerical weather prediction models as well as for climate prediction (Rowntree, 1991). It is also of prime importance for catchment scale hydrology as a major component of the terrestrial water cycle, especially over semi-arid regions. It is, finally, a key variable in agronomy for irrigation scheduling. However, it is also recognized as one of the most uncertain

components of the hydro-climatic system (Jasechko et al., 2013). In semi-arid regions of the southern Mediterranean, the agriculture consumes about 85% of the total available water and is on continuous expansion (Voltz et al., 2018). With an efficiency lower than 50% due to the use of the traditional flooding systems and to the poor scheduling of irrigation, pushing forward our knowledge of the ET and its partition is also of prime importance for improving the management of agricultural water in this region.

Soil moisture patterns and in particular the spatial gradients have been found to impact the development of convective storms through changes of the thermo-hydric characteristics of the low atmosphere (Koster et al., 2004; Taylor et al., 2011). In




semi-arid regions, irrigation causes contrasting soil moisture conditions and cools and moistens the surface over and downwind of irrigated areas (Lawston et al., 2015). Precipitation may be enhanced downwind (DeAngelis et al., 2010) while it could be slightly reduced over the irrigated areas, likely as a result of a reduction in both local convection and large-scale

moisture convergence (Pei et al., 2016). Irrigation also drastically affects the partition of available energy into sensible and latent heat fluxes (Ozdogan et al., 2010), promotes sensible heat advection from the surrounding drier surface (Lei and Yang, 2010) and impacts the partition of ET into plant transpiration $T_r$, usually associated with plant productivity, and soil evaporation E that is lost for the plant (Kool et al., 2014). In this context, Hartmann (2016) suggests that transpiration may be more efficient than bare soil evaporation in enhancing the land-atmosphere feedbacks. Indeed, transpiration is associated

to longer climate memory than soil evaporation as plant roots can extract water from a deep reservoir and maintains a regular input of water to the atmospheric boundary layer while the small evaporative layer of soils dries out in several days, in particular on semi-arid regions.

Within this context, the micro-meteorological community has developed numerous Soil-Vegetation-Atmosphere Transfer scheme (SVATs) with varying degrees of complexity to estimate ET and its partition (Noilhan and Planton, 1989; Sellers et

al., 1996; Noilhan and Mahfouf, 1996; Coudert et al., 2006; Gentine et al., 2007). In parallel, several studies have examined the representation of surface heterogeneity by SVATs and in particular concerning the surface energy budget. Part of the existing SVATs generally solve a single composite energy balance for the soil and the vegetation and thus calculate a composite temperature. These "mono-source" models have been used successfully on herbaceous, dense and homogenous covers (Kalma and Jupp 1990, Raupach and Finnigan 1988). By contrast, they may not be suited for sparse vegetation (Van

Hurk et al., 1995; Blyth and Harding 1995; Boulet et al., 1999) that is a common feature of south Mediterranean crops. Indeed, these covers are characterized by a high heterogeneity in terms of geometry (rank, several layers) and hydric status, especially for tree crops. In the case of irrigated sparse cover, the temperature contrast can be high between, on one hand, a dry and hot soil interacting directly with the atmosphere and receiving a large fraction of incoming radiation not screened by the vegetation, and, on the other hand, a well-watered vegetation transpiring at its potential rate thanks to irrigation. In

addition, the heat sources composing complex crops (soil, tree cover, potential intermediate annual cover …) such as trees are coupled to varying degrees depending on the heterogeneity of the crop. The representation of the intensity of this coupling, and ultimately the performance of the models to reproduce the ET and its partition, is directly related to the structure adopted in the model (single- or dual- source). In particular, it has been shown that a more realistic representation of the energy balance and a better representation of the respective contributions of E and $T_r$ to ET (Shuttleworth and

Wallace, 1985; Norman et al., 1995; Béziat et al., 2013; Boulet et al., 2015) could be obtained by solving several separate energy balances for each of the sources. In this context, two types of dual-source models were developed (Lhomme et al., 2012). The coupled or layer approach considers that the canopy is located above the soil layer (Shuttleworth and Wallace, 1985; Shuttleworth and Gurney, 1990, Lhomme et al. 1994, 1997) while for the uncoupled or patch approach, soil and vegetation sources are located next to each other. This means that, for the layer representation, exchanges of heat and



moisture between the soil and the atmosphere go necessarily through the vegetation layer as it covers completely the ground.
By contrast, for the patch representation, soil and vegetation turbulent processes are independent and soil receives the full
incoming radiation not screened by the vegetation (Norman et al., 1995; Kustas and Norman, 1997; Boulet et al., 2015). The
choice between the patch and the layer approach is related to the scale of the surface heterogeneity (Lhomme and
Chehbouni, 1994; Boulet et al., 1999; Lhomme et al., 1999; Blyth and Harding, 1995; Lhomme et al., 2012). Roughly, a

layer approach should be adopted if the scale of heterogeneity is small while the uncoupled representation is better suited for
larger patches allowing for uncoupled surface boundary layers above each patch. The ratio of vegetation height to the patch
size has been proposed as indicator of canopy heterogeneity. Blyth and Harding (1995) and Blyth et al. (1999) found that the
coupled model represented better the data in the extreme case of a tiger bush characterized by a ratio of 1/10 than the patch
approach.  By contrast, Boulet et al. (1999) highlighted that the patch approach was more realistic to predict the energy

balance of sparse but relatively homogeneous area dominated by shrub and bushes in the San Pedro Basin. The question thus
arises of what is the threshold for choosing one representation from the other? The question is particularly relevant for
complex tree crops in the Mediterranean areas such as Olive orchard because a large diversity of field geometry co-exist in
the Mediterranean area from the sparser rainfed fields to the denser intensively cropped fields with new tree varieties.
Finally, another modeling issue for irrigated agro-system is the highly transient soil moisture regime induced by irrigation

and the strong energy switch between latent and sensible heat fluxes at the irrigation time.

The Interaction Soil Biosphere Atmosphere (ISBA) model is part of the SURFace EXternalisée platform (SURFEX) from
Météo-France (Masson et al., 2013). It provides the land surface limit conditions for all the atmospheric models of Météo-
France and is used in the operational hydrological system (named SIM for SAFRAN-ISBA-MODCOU; Habets et al., 2008).
The standard version of this model (Noilhan and Planton, 1989) uses a single composite soil-vegetation surface energy

budget meaning that only a composite soil-vegetation temperature is solved by the model (Noilhan and Planton, 1989;
Noilhan and Mahfouf, 1996). Recently, Boone et al. (2017) have developed a multiple energy balance (ISBA-MEB) version
that can represent the surface with up to three sources including the snow layer as there are big issues to improve the
representation of the snowpack effect on surface temperature for northern latitude forest ecosystems. This new version of
ISBA gives a unique opportunity to compare single and dual-source representations of irrigated crops, including complex

tree crops, within the same modelling environment (meaning that all other processes are parameterized in the same way). It
was evaluated on temperate forested areas (Napoly et al., 2017) without investigating the partition of evapotranspiration.

The main objective of this study is to evaluate the added value of the multiple energy balance in ISBA/SURFEX to simulate
surface heat fluxes and the partition of ET into $T_r$ and E over two dominant crop types in the Mediterranean region which are
irrigated using the traditional flooding technique. This paper is organized as follows: i) description of the experimental sites

and data; ii) description of the model versions and their implementation; iii) sensitivity analysis and model calibration; iiii)
comparison of the different ISBA model representation and discussions.



## 2. Data and Land Surface Model ISBA-A-gs

### 2.1 Study sites and *in situ* measurements

#### 2.1.1 Study region

The region of study is the Haouz plain located in the Tensift basin (Marrakech, Morocco; Figure 1). The climate of the area is similar to that of the semi-arid Mediterranean zones with hot and dry summers and low precipitation which mostly falls between November and April of each year. The annual average ranges between 192 mm and 253 mm per year, largely lower than the evaporative demand which is around 1600 mm/year (Jarlan et al., 2015; Chehbouni et al., 2008). In this region, the dominant irrigated crop including arboriculture (olives and oranges) and cereals (wheat) consumes about 85% of available

water which comes from groundwater pumping or dams. As reported in Ezzahar et al. (2007), the majority of the farmers (more than 85%) use the traditional flood irrigation method which causes much loss of water through deeper percolation and soil evaporation. In this study, two flood-irrigated sites of olive orchard and winter wheat have been instrumented with micrometeorological observations.

#### 2.1.2 The olive orchard site

An experiment was set up in an olive orchard site (31°36'N, 07°59'W) named "Agdal" located in the vicinity of Marrakech city during the 2003 and 2004 growing seasons (Figure 1). The site occupies approximately 275 ha of olive with an average height of about 6.5m and a density of 225 trees/ha. The irrigation water are collected after snow melting and stored into two basins. Afterwards, a ditch network is used to divert water from basins to each tree which is surrounded by a small earthen levy. The latter retains irrigation water needed for each tree (Williams et al., 2004). Depending on available manpower, the

irrigation of the total area takes approximately 12 days. For more details about the description of the Agdal site and related experimental set-up, the reader can refer to Ezzahar et al. (2007, 2009a and 2009b).

#### 2.1.3 The winter wheat site

The second experiment was carried out in the irrigated perimeter named "R3" (Figure 1), situated thereabouts 45 km east of Marrakech city (31°68'N, 7°38'W). R3 is about 2800ha and the main crop is flood-irrigated winter wheat. Depending on the

first heavy rainfall during the winter season and climatic conditions, the wheat is generally sown between November and January, and harvested in the end of May. Based on the dam water level at the beginning of each agricultural season, the amount of irrigation water and frequency are managed by the Regional Office of Agricultural Development of the Haouz plain (ORMVAH). Two wheat fields were instrumented during the seasons 2002-2003 and 2012-2013. The site description and experimental set-up details are more provided in Duchemin et al., 2006, Er-Raki et al., 2007, Le page et al., 2014 and

Jarlan et al., 2015.



### 2.1.4 Data description

Meteorological and micro-meteorological data

A meteorological station was installed over each site to measure air temperature and humidity, wind speed and direction, incoming solar radiation and rainfall. Likewise, net radiation and its components, soil temperature and soil water content at

several depths and soil heat flux were also measured. All measurements were collected on half-hourly basis. Sensible and latent heat fluxes were measured using an eddy covariance method which consisted of a 3D sonic anemometer and krypton hygrometer that measures the fluctuations of the three components of the wind speed, air temperature and water vapor. Measurements were taken at high frequency (20 Hz) and stored on a CR 5000 datalogger using a PCMCIA card. These measurements were collected and processed by an eddy-covariance software "ECpack" in order to derive sensible and latent

heat fluxes by including all corrections reported in Hoedjes et al. 2007.

Evapotranspiration partition

In addition to the EC observations, two techniques were used to measure separately the plant transpiration and the soil evaporation:

(1) Isotopes observations: The stable isotopes tracer technique was applied for the R3 site. This technique measures the isotopic compositions of Oxygen ($\delta^{18}O$) and Hydrogen ($\delta^2H$) of water fluxes from the soil water and foliage and quantifies the rate of the plant transpiration and soil evapotranspiration to the total evapotranspiration (ET). The sampling of soil, atmospheric and vegetation water samples were made during two days (Day Of Year -DOY- 101 and 102) of the growing season 2012-2013 and were analysed for their stable isotopic compositions of $\delta^{18}O$ and $\delta^2H$. It should be noted that the

sampling was made during the development stage with cover fraction larger than 0.8. Also, the soil was very dry with a soil moisture of about 0.12 $m^3/m^3$ because the experiment was conducted before an irrigation event which was applied on DOY 104. Atmospheric water vapour was sampled from four heights (0cm, 85cm, 2m and 3m), between 10:00 and 16:00h with a frequency of 1hour on each sampling day. In addition, the samples of soil and vegetation were collected approximately between 13h and 14h. Afterwards, these samples were used to calculate $\delta^2H$ of the soil, vegetation and atmosphere in order

to estimate the ET partition based on the Keeling plot approach and then to compare it with the modelled soil evaporation and plant transpiration. More details about the description of the principles and techniques of observations can be found in Aouade et al. (2016). (2) Sapflow observations: Similarly to R3 site, Heat Ratio Method (HRM) was applied for Agdal site to measure xylem sap flux of eight olive trees using heat-pulse sensors. The period of measurement was situated between 9th of May (DOY 130) and the 28th of September (DOY 272) during 2004. This period is characterized by a hot climate with

very high surface temperatures and thus presents a perfect period for studying the ET partition over such surfaces. In brief, this method uses temperature probes which were inserted into the active xylem at equal distances upstream and downstream from the heat source. This method was chosen due to its high precision at low sap velocities and its robust estimation of transpiration of olive (Fernandez et al., 2001). The heat-pulse sensors were equally inserted into large single and multi-




stemmed trees located in the vicinity of the EC tower. The transpiration at the field scale (in mm.day$^{-1}$) was obtained by

scaling the measured volumetric sap flow (L.day$^{-1}$) based on a survey of the average ground area of each tree (45 m²). Finally, based on the EC observations, the obtained single tree transpiration was extrapolated to the EC footprint scale which is representative for the whole field (Williams et al., 2004 and Er-Raki et al., 2010).

Vegetation characteristics and irrigation inputs

The mean fraction cover ($F_c$) and the leaf area index (LAI) obtained from one campaign of hemispherical canopy photographs (using a Nikon Coolpix 950 digital camera fitted with a fisheye lens converter 'FC-E8', field of view 183°) are equal to 55% and 3 m²/m², respectively for the Olive site. For the wheat site, $F_c$ and LAI together with vegetation height $h_c$ were measured about every 15 days using the same instrument. Irrigation dates and amount were also gathered by dedicated surveys. Time series of LAI and reference evapotranspiration $ET_0$ are provided as supplementary material (Figure S1).

**2.2  The ISBA-A-gs model description and implementation**

**2.2.1    Model description**

ISBA is a land surface model used to simulate the heat, mass, momentum, and carbon exchanges between the continental surface (including vegetation and snow) and the atmosphere. It also prognoses temperature and moisture vertical profile in the soil. The first developed version of the ISBA model named thereafter as "standard version" based on a simple soil-

vegetation composite scheme to compute the surface energy budget was developed by Noilhan and Planton (1989) and Noilhan and Mahfouf (1996). It is implemented within the open-access "Surface Externalisée" (SURFEX) platform version 8.1 developed at CNRM/Météo-France (Masson et al., 2013). In this study, a multilayer soil diffusion scheme (Decharme et al., 2011) is used to simulate the soil water and heat transfers instead of the initial force-restore formulation (Deardoff, 1977). The soil is vertically discretized by default into 14 soil layers up to 12 m depth to ensure a realistic description of the

soil temperature profile (Decharme et al., 2013) while water transfers are active on the 0-2m layer only (Decharme et al., 2013). Moisture and temperature of each layer is then computed according to their textural and hydrological characteristics. The latters (hydraulic conductivity and soil matrix potential) are derived from the Brook and Corey (1966) parameterization following Decharme et al. (2013). The stomatal conductance and the photosynthesis are computed using the $CO_2$-responsive parameterization named A-$g_s$ (Calvet et al., 1998, 2004). The model includes two plant responses to soil water stress

functions depending on the plant strategy with regards to drought (Calvet, 2000; Calvet et al., 2004). Non-interactive vegetation option is chosen meaning that vegetation characteristics (LAI, height and fraction cover) are prescribed from in situ measurements with a 10-days time step. The multi-layer solar radiation transfer scheme (Carrer et al., 2013) which considers sunlit and shaded leaves is also activated. The root density profile is a combination of an homogeneous profile and of the Jackson et al. (1996) exponential profile (Garrigues et al., 2018). Full expressions of the aerodynamic resistances are

given on Noilhan and Mahfouf (1996).





Compared to the standard version of ISBA-A-g$_s$, ISBA-MEB, for Multiple Energy Balance, solves up three separated energy budgets for the soil and the snowpack following Choudhury and Monteith (1988). In this study, a double source arising from the soil and from the vegetation is used. For extended details about the different hypothesis used in MEB version as well as its full mathematical formulas and its related numerical resolution methods, the reader can refer to Boone et al. (2017). The

main governing equations of both versions of the model are given in appendix 1.

### 2.2.2    Model implementation

Input parameters and data

ISBA within SURFEX is intended to be implemented using the patch approach where each grid point can include up to 19 patches representing 16 different plant functional type, bare soil, rock and permanent snow. Within the SURFEX platform,

input parameters and variables are usually derived from the ECOCLIMAP II data base (Faroux et al., 2013). In this study, ECOCLIMAP II is bypassed by using in situ measurements for most of vegetation characteristics and albedo. For the wheat site, 10-days vegetation characteristics (LAI, h$_c$ and F$_c$) were derived from in situ measurements based on a linear interpolation. Annual constant values were used for the Olive orchard. The roughness length for heat and momentum exchanges (Z$_{0m}$ and Z$_{0h}$, respectively) are derived from h$_c$ following Garrat (1992): Z$_{0m}$=h$_c$/8 and Z$_{0m}$/Z$_{0h}$=7. The emissivity

and the total albedo are obtained as a linear combination of the soil and vegetation characteristics weighted by the fraction cover. The total albedo derived from the two components of the short wave net radiation measured by the net radiometer (CNR1) instruments are used to calibrate the albedos of vegetation and soil for the whole study field. The two component albedos remain constant for the whole set of simulations while the total albedo evolves through the vegetation cover fraction changes. Input data for the two sites are summarized in Table 1. Soil hydraulic properties were computed from the Clapp and

Hornberger (Clapp and Hornberger, 1978) and the Cosby et al. (1984) pedotransfer functions. The resulting parameters were quite similar: W$_{wilt}$=0.25 and W$_{fc}$=0.34 for Clapp and Hornberger and W$_{wilt}$=0.26 and W$_{fc}$=0.33 for Cosby et al. (1984) Nevertheless, values obtained based on the calibration on soil moisture time series were quite different (W$_{wilt}$=0.18 and W$_{fc}$=0.41). Beyond the inherent uncertainties of the pedotransfer functions, this may be mostly explained by the lack of representativity of the soil sampling. Calibrated W$_{wilt}$ and W$_{fc}$ were imposed.


Model configurations

Three structural representations of the canopy are compared in this work: (1) the composite energy balance of the standard version named afterwards ″ISBA-1P″ for the single patch version; (2) the uncoupled version noted ″ISBA-2P″ for two patches, where the canopy and the soil patches are situated side-by-side, resolves two energy balance equations for both

patches without any interactions concerning the turbulent heat exchanges. Likewise, the soil water dynamic is predicted on two uncoupled soil columns; (3) The coupled two layer approach of the new MEB version ″ISBA-MEB″ where the canopy





layer is located above the soil component and the energy budgets of both layers are implicitly coupled with each other (Boone et al., 2017). Note than the ISBA-2P configuration is implemented on the Olive orchard only as there is no reason to represent the homogeneous canopy of wheat crops by two patches located side by side. Figure 2 displays the schematic

representation of the 3 configurations of the model.

## 2.3 Sensitivity analysis and parameters calibration

### 2.3.1 Sensitivity analysis and calibration methods

Analyzing the sensitivity of the parameters one by one is not satisfactory because of the parameter interactions and non-linearities in the model equations and in the underlying processes (Pianosi et al., 2014). For this reason, the multi-objective

generalized sensitivity analysis (MOGSA) (Goldberg, 1989; Demarty et al., 2005) is chosen in this study. The MOGSA methodology uses a Monte Carlo sampling of the search space. To represent the uncertainty of parameter estimates, an ensemble of N parameter set is drawn stochastically within a range of physically realistic values using an uniform distribution. A threshold on the targeted objective functions is then used to partition the ensemble into an "acceptable" and a "non-acceptable" regions. The trade-off between the targeted objectives is sought using a Pareto ranking scheme. The

cumulative distribution of parameters value is compared to the normal distribution through the statistical Kolmogorov-Smirnorff (KS) test that relates this maximal distance to a probability value. The application of thresholds to this probability value permits to quantify the degree of parameter sensitivity. Ensemble of 20000 simulations for Agdal and 40000 for the R3 sites were computed. The size of simulations is related to the size of the studied period. Based on the recommendations of Demarty (2001), it is assumed that the size of the samples was large enough to obtain robust results. No account was taken of

possible covariation between the parameter values in these prior choices of parameter sets because such covariation is generally difficult to assess. Several couple of objective functions was explored: latent heat LE and transpiration $T_r$, and sensible heat H and $T_r$, and LE/H. As similar sensitive parameters were highlighted, the chosen objective functions in this work were the convective fluxes H and LE. The MOGSA algorithm is also used to retrieve the parameter set providing the best trade-off of objective functions (Demarty et al., 2005). This parameter set will be called hereafter "optimal". The ISBA

model was thus calibrated by taking the best parameter set among the 20000 and the 40000 tested in the multi-objective sense. Finally, the validation step was carried out over the 2004 and 2013 seasons for Agdal and R3, respectively.

### 2.3.2 Sensitive parameters selection

For our sensitivity study, a total of 16 parameters ($\phi_v$ and $\phi'_v$ are for MEB only) were identified based on a previous knowledge of the model and the rich literature based on the use of ISBA-A-g$_s$ and ISBA-MEB (Calvet et al., 2001, 2008;

Boone et al., 1999, 2009, 2017; Napoly et al., 2017). The list of parameters and their ranges of variation are reported in Table 2. The land cover database obtained from ECOCLIMAP (Masson et al., 2003; Roujean et al., 2013) and ECOCLIMAP-II (Faroux et al., 2013) were used to prescribe the range of variations of the input parameters. The same





sensitivity analysis and calibration study were conducted for the standard single source version and the MEB version of ISBA. The sensitivity analysis was carried out for the whole 2003 wheat season for the R3 site and between 1st of June and

30 August (2003) over the olive orchard (Agdal site) in order to limit the computing time.

The parameters list includes : (1) some well known to be highly sensitive parameters such as the soil texture, the root depth and the ratio of roughness lengths $Z_0/Z_{0H}$; (2) some parameters of the A-$g_s$ module: the mesophyllian conductance in unstressed conditions $g_m$, the maximum air saturation deficit $D_{max}$, the cuticular conductance $g_c$ and the critical normalized soil water content for stress parameterization $\theta_c$ (Calvet et al., 2000; Calvet et al., 2004; Rivalland et al., 2005);

(3) the new parameters which were introduced in ISBA-MEB , such as the longwave radiation transmission factor, which determines the partition of this radiation between vegetation and soil (Boone et al., 2017) and the attenuation coefficient for momentum and for wind that prescribe changes based on canopy heights, turbulent transfer coefficients, and wind speed (Boone et al., 2017, Choudhury and Monteith, 1988). Values of these two parameters are, in the current version of ISBA, constant independently of the type of canopy ($\phi_v = 2$ and $\phi'_v = 3$) while Choudhury and Montheith (1987) have shown that

this model is sensitive to the variation of those two parameters, in particular, the temperature of the ground surface, which depends, among other things, on the aerodynamic resistance between the source of movement at the vegetation level and the soil surface. Likewise, the aerodynamic resistance between the vegetation and the air at the vegetation level is related to $\phi'_v$ and to the Leaf width $L_w$ (Choudhury and Monteith, 1988).

All parameters are common between the two versions except $\phi_v$, $\phi'_v$ and $L_w$ which concern the MEB version only.

## 3    Results

### 3.1 Sensitivity analysis and calibration

Only results of ISBA-MEB are presented here as quite similar list of sensitive parameters is obtained with the standard version of ISBA. The simulations are partitioned into two groups: "acceptable" and "unacceptable". Demarty et al. (2005) suggested that 7 to 10% of members should compose the "acceptable" set. In this context, 1720 acceptable simulations for

the Agdal site (8.6%) and 3600 for the wheat site (9.0%) are retained. Figure 3 displays the results of the sensitivity analysis obtained for both sites. The horizontal dashed lines indicate the transition levels between 'low', 'medium' and 'high' sensitivity (Bastidas et al., 1999). Table 3 reports the optimal values of the highly sensitive parameters for at least one of the objective functions.

The high sensitivity of some parameters was anticipated such as: (1) the soil texture related parameters (fraction of sand

and/or clay) that strongly impacts the hydrodynamic characteristics of the soil and, ultimately, the fluxes (Garrigues et al., 2015); (2) the root depth that has a   major  role in the extraction of available water in the root zone (Calvet et al., 2008); (3)





the ratio of roughness lengths $Z_0/Z_{0H}$, which impacts the calculation of the aerodynamic resistance. Those parameters which highly affect the model behavior are usually estimated through *in situ* measurements or for a large-scale application from global data base. In both case, their values are uncertain, even at the station scale as their spatial variability remains

significant, including the soil texture along the vertical profile. Five other sensitive parameters are also common to both sites in particular the long wave transmission factor $\tau_{LW}$ introduced in the new radiative transfer scheme and the parameters

introduced in the ISBA-MEB version $L_w$, $\phi_v$, $\phi'_v$ and $U_l$. Concerning the attenuation coefficient of the movement $\phi_v$, Choudhury and Montheith (1988) had already shown the strong sensitivity of the model to this parameter especially for dry soils encountered in our study sites.


Concerning the Agdal site, results showed 11 sensitive parameters, 8 parameters with 'high' sensitivity and 3 with 'medium' sensitivity (Figure 3a) when at least one of the objective functions are considered. The chosen period was characterized by a gradual drying of the soil with a water stress detected on day 190 (Ezzahar et al., 2007). The plant transpiration thus represented the main component of the evapotranspiration. Within this context, the identified sensitivity of parameters

directly impacting the stomatal regulation ( $g_m$, $D_{\max}$ ) and the availability of water in the soil (Sand, Clay and RD) is consistent. Regarding the moderate fraction cover ($F_c$=0.55) and the flooding technic applied for irrigation, soil evaporation tightly related to soil texture as well may not be negligible on the site. Interestingly enough, the obtained optimal values of 0.47 for Sand and 0.27 for Clay (Table 3) were very close to the *in situ* measurements. The root depth (RD) influences also strongly convective fluxes. An optimal value of 0.62m. was found while literature as well as ECOCLIMAP, propose deeper

rooting depth up to 1.5m. for perennial trees. Nevertheless, it is well known that roots develop in the upper wet layer of the soil when irrigation is applied (Fernandez et al., 1990) while deeper development can be observed in case of water supply problems only (Maillar, 1975). Additionally, the soil in our site below 1 m. is very compact and contains rocks which limit the development of the pivoting roots.

For the wheat site, the sensitivity analysis revealed 13 sensitive parameters: 8 of them have a 'high' sensitivity and 5 of them with a 'medium' sensitivity (Figure 3b). As for the Agdal site, specific parameters are related to the soil (like Cl) and others related to the crop (RD, $L_w$ and $\tau_{LW}$ ). By contrast to Agdal, the two fluxes LE and H showed also a strong sensitivity to the $Z_0/Z_{0H}$ parameter. In the standard version of ISBA-A-g$_s$, this ratio is equal to 10 according to Braud et al. (1995) and Giordani et al. (1996), but at the station scale, several studies have shown that this ratio could range from 1 to 100 (Napoly et

al., 2016). The optimal value for the wheat site was 7.00. The obtained optimal lower value increases the amplitude of H and reduces that of the surface temperature. This is consistent with similar findings of Beziat et al. (2013) for a wheat site located in the South-West of France. Literature as well as ECOCLIMAP, propose values of root depths of about 0.50 m for our type of crop (Crop C3). In our case, a slightly higher value of 0.55 m. appeared optimal for latent heat fluxes and also for





transpiration when compared to the isotopic measurements (see comments below and Table 3). This is an acceptable value

for irrigated wheat in the region (Duchemin et al., 2006, Er-Raki et al., 2007). Due to the limited number of irrigations on this site, the plant tends to extend roots to deeper layers to extract water. The slightly lower value of clay content (0.44) than the *in situ* measurement (0.47) adjustment is also consistent by limiting water retention and favors water availability in the deepest layers. Regarding the evapotranspiration flux, in the case of a dry soil, the only possible solution to reproduce the experimental data is to increase transpiration of the crop. Indeed, the appearance of a strong sensitivity of the two parameters

$\phi_v$ and $\phi'_v$, seems to be consistent (Choudhury and Idso 1985) and optimal higher (lower) values than literature of $\phi_v$

($\phi'_v$) are obtained (Table 3).

As a conclusion, the high sensitivity to the new parameters introduced in ISBA-MEB and the optimal values different from default means that studies at the local scales should be duplicated for different eco- and agro-systems to feed the ECOCLIMAP-II data base with specific parameter values in the view of a large-scale applications.

**3.2   Composite energy budget**

**3.2.1    Latent heat flux**

Figure 4 displays the daily time series of latent heat fluxes using the three configurations of the ISBA model for both sites. The irrigation and rainfall events are also superimposed. The same figure but for the sensible heat flux is provided as supplementary material (Figure S2). Statistical metrics for the four components of the energy budget are reported in Table 4.

The seasonal dynamic of LE is properly reproduced by the three configurations of the model for both sites. ISBA-2P and ISBA-MEB definitely outperformed the ISBA-1P version on average over the olive orchard for both seasons with RMSE values below 52.2 W/m². while for ISBA-1P, RMSE can reach up to 107.1 W/m². By contrast, over the wheat site, the ISBA-1P version is much closer to ISBA-MEB with differences of RMSE below 10 W/m². This means that: (1) the dual source configurations are better suited to predict composite LE for row crops of moderate fraction cover while, as expected, a

simple composite energy budget can cope with the homogeneity of the wheat canopy at least to predict LE; (2) the slightly better results obtained with ISBA-MEB than with ISBA-2P demonstrates that the soil and the vegetation heat sources are coupled to some extent. This is probably because the bare soil area between the tree rows is not sufficiently large to consider two independent heat sources located side by side. Previous studies have already demonstrated the limit of single source models for predicting surface fluxes over sparse vegetation. Jiménez et al. (2011) have evaluated four single source (Mosaic,

Noah, Community Land Model -CLM-, and Variable Infiltration Capacity -VIC-) at the global scale and they showed their limitations for producing latent and sensible heat fluxes over tall and sparse vegetation such as forest canopies. Likewise, Blyth et al. (1999) over Sahel estimated more accurately the surface fluxes over Savannah with the dual source version of the MOSES model compared to the original single source version. Our results with the new ISBA-MEB version implemented within the SURFEX platform are consistent with these previous findings.






Another interesting feature is the observed departure between model predictions and observations around irrigation events. It is well known that part of the observed discrepancies between simulations and observations can be related to the eddy covariance measurements because of the associated strong heterogeneity within the footprint during an irrigation event. Nevertheless, the different configurations of the model strongly differ during these specific periods, in particular over the

Olive site. In line with the observations, the three configurations show a strong shift of the available energy from sensible to latent heat when irrigation occurs (see also sensible heat flux time series, figure S2) but, while this shift is moderate and in overall agreement with the observations for the dual source configurations, it is strongly emphasized by ISBA-1P. For instance, LE predictions reached a maximum of about 550 W/m² in mid-June for both seasons 2003 and 2004, observations remained below 400 W/m². To a lesser extent, this trend to unreasonable shifting also occurred for the ISBA-2P especially

when the available energy is very high (during the summer months of the 2003 season). The reverse behavior is obviously observed for H (Figure S2): after each irrigation event, the simulated sensible heat by ISBA-1P dropped considerably due to the drastic decline in simulated surface temperature by this version.

By contrast, on the wheat site, the dynamic of the latent heat flux is smoother than at the Olive site in 2013 and, to a lesser

extent in 2003, in particular because of a persistent cloud cover during the first two weeks of March (cf. the drastic drop of $ET_0$, Figure S1). The year 2003 is also characterized by lower LE values mainly because several successive drought years in the beginning of the 2000s cause a drop of dam levels and limited the water availability for irrigation. Indeed, the total cumulated rainfall and irrigation was 351mm for the 2002-2003 season while it reached about 770mm for 2012-2013. By contrast to the Olive site, the two configuration of the model are able to reproduce the overall seasonal dynamic of LE for

these two contrasted years. The only exception is around the late season irrigation events in April and May for year 2013 during which ISBA-1P showed the same trend to strongly emphasize the energy shift as already highlighted for the Olive site.

As a conclusion, while dual sources configuration outperformed the single source version of the model for the complex and sparse Olive canopy, a composite single energy budget is able to reproduce the seasonal dynamic of LE for the homogenous

wheat cover. At this level of sparsity for the Olive orchard, the coupling between soil and vegetation heat source is moderate as both the patch uncoupled and the coupled layer configurations provided close statistical metrics. For the Olive site, significant drawback of ISBA-1P and, to a lesser extent, ISBA-2P is highlighted during the strong transient regime associated to the irrigation events.

### 3.2.2 Other components of the energy budget

The performance of the different configurations to simulate the other components of the energy budget $R_n$, H and G was investigated using a Taylor diagram (Figure 5). This presentation summarizes graphically the comparison between the model





and the observations based on root mean square difference, correlation coefficient rand standard deviations (Taylor, 2001). Statistical metrics are reported in Table 4.

The net radiation is almost perfectly simulated by the three configurations with slight differences related to the budget in the
longwave. Values of the albedo are identical for the three configurations and have been calibrated on the short wave components of $R_n$ measured by CNR1. For both sites, the correlation coefficient r is close to 1.0 and the RMSE is lower than 25.0 W/m². These good performances are in agreement with results reported in the literature. Indeed, several studies showed that the estimation of $R_n$ by SVAT models is good on several type of canopy (Napoly et al., 2017, Boulet et al., 2015, Ezzahar et al., 2007 and 2009a). The most important differences are encountered on the Agdal site and can be explained by
the slight overestimation (not shown) of the infrared radiation ($LW_{up}$) by ISBA-2P for both seasons (Bias=13.7 and 12.7 $W/m^2$ for the 2003 and 2004 seasons, respectively). For this configuration, the soil directly exposed to the solar radiation becomes very hot and dissipates much less energy by conduction compared to the other two configurations (cf. below).

As for the latent heat flux, the dual sources configurations outperformed the single source version for sensible heat flux
predictions over complex cover with wide differences in performances between ISBA-1P on one hand, and ISBA-2P and ISBA-MEB on the other hand over the olive orchard site (Figure 5). By contrast, the two tested configurations are much closer for the wheat site with RMSE of 55.2 and 55.0 $W/m^2$ in 2003 for ISBA-1P and ISBA-MEB, respectively. The temporal dynamic is also greatly improved since the correlation coefficients (r) are above 0.8 for the dual source configurations for the 2 seasons at the olive orchard site whereas they are only of 0.7 and 0.6 for the simple source approach.
Here again, it is the shifting between sensible and latent heat fluxes during the irrigation events and during the drying period which leads to the differences between single and dual sources configurations (see also figure S2). For the wheat site, the behaviors are very similar for the two configurations although the MEB version presents the best performances.

Due to the complexity of the canopy surface and the spatial variability of hydric and thermal conditions, particularly because
of the shading effects, the ground heat flux is the most difficult component of the energy budget both to simulate and to measure. The heat plates fluxes used on both studied sites have a very low representativeness which does not exceed a few tens of centimeters whereas the illumination can be very variable under a relatively open canopy such as olive or over wheat at the beginning of the season. Therefore, the obtained results should be interpreted with caution. The highlighted improvement on the convective fluxes using –MEB with regards to the two other configurations is not so clear for the
conduction fluxes. It seems that MEB has a systematic tendency to dissipate too much energy by conduction (cf. biases on Table 4). On average on the two years for the Olive site, the ISBA-2P configuration has the best overall performance in predicting G. Nevertheless, by taking a closer look on the daily cycles of the ground heat flux with a distinction between the bare soil patch and the vegetation patch (cf. supplementary material; Figure S3), it appears that the patch bare soil dissipates a lot of energy by conduction as shown by the amplitude of the daily cycles which is much stronger than observations. By
contrast, the patch vegetation (LAI =5 and a cover fraction close to 1) dissipates less energy by conduction in favor of the





convective fluxes. The average ground heat flux (derived as the sum of the two components weighted by their respective fraction cover) is in good agreement with observations even if none of the 2 patches represent correctly the observed fluxes. This tends to show that the uncoupled approach is quite suitable for predicting total G over this sparse and relatively open cover but for the "wrong" reasons.

### 450  3.3  Soil and Vegetation components

In this section, soil and vegetation components of the radiation budget and of the partition of evapotranspiration were analysed. Please note that only the olive orchard site was considered for the radiation budget components as the experimental design on the wheat site could not sample each component separately. In addition, ISBA-MEB only is able to predict the radiation budget components. For ISBA-2P, vegetation and soil refers to the components predictions of the respective patch 455  while for ISBA-1P, the soil-vegetation composite variables are plotted.

#### 3.3.1  Radiation budget

Figure 6a displays the time series of the soil net radiation simulated by the three configurations at the Agdal site in 2003 season. Similar conclusions can be drawn from the data acquired in 2004. The ''bare soil'' patch is shown for the ISBA-2P configuration. The most striking feature is the significantly higher energy available for convection and conduction at the soil 460  level for ISBA-1P and ISBA-2P with regards to ISBA-MEB (the reverse is obviously true for vegetation net radiation, not shown). The amplitude of the seasonal cycle is much stronger for the observations than for the ISBA simulations. ISBA-MEB is in better agreement with observations during the winter months while the agreement is better in summer for ISBA-2P and ISBA-1P. The strong differences may be related to observations. Indeed, the soil net radiation was measured under cover. When the cover is sparse, as for the olive trees, it is very difficult to screen it totally from direct incoming radiation as 465  during summer, with a solar zenith angle close to 0, the instrument is exposed to direct radiation. In winter, ISBA-MEB appears to be well reproducing the measurements of the available energy at the ground-level when the instrument may be shadowed by the canopy. By contrast, when the instrument is exposed to direct illumination, the ISBA-1P configuration and the bare-soil patch of the ISBA-2P configuration are obviously closer to observations.

Figure 6b displays the time series of the soil temperature at the Agdal site in 2003. The new coupled version limits the available energy arriving at the ground level compared to the single-source version and therefore leads to the lower predicted temperatures. The patch bare soil of the ISBA-2P configuration exhibits the higher values of soil temperature because it is directly exposed to incoming solar radiation. ISBA-1P lies in-between because the canopy is open ($F_c$=55%). On average, biases for ISBA-MEB and ISBA-1P are moderate but it is due to an overestimation by both configurations during winter 475  while an underestimation is observed during summer. Indeed, for the winter months, the temperature sensor observes mainly areas of shaded bare ground while, during summer, the observed soil is under the influence of direct illumination. At this time, the bare soil patch of the ISBA-2P configuration presents the best agreement with the observations (Bias= -2.3° for



June to September compared to -6.6° and -6.7° for ISBA-1P and ISBA-MEB, respectively). Moreover, this negative bias is mainly attributed to the few days following the irrigation events for which the bare soil patch simulates a much greater cooling. The difference reaches more than 7.0°, three days after the irrigation at the beginning of August in particular. On the other hand, when the soil is dry (more than 10 days after each irrigation), the difference is less than 1.5°. There is also a fairly clear over-estimation during the winter months. At this time of the year, H is slightly underestimated.

Finally, figure 6c is the same as figure 6b but for vegetation temperature. These observations may be more reliable than the observations of the soil temperature even if some parts of the bare soil can disturb the representativeness of the observations. The three configurations are much closer than for the observations of soil temperature and reproduce reasonably well the observations with RMSEs of 4.5°, 4.6° and 4.2° for ISBA-1P, ISBA-2P and ISBA-MEB, respectively. A large part of these errors can be attributed to the positive bias of the three configurations. The ISBA-MEB version has the lowest bias while the ISBA-1P and ISBA-2P versions are logically slightly warmer. Indeed, ISBA-MEB is able to partition the energy between the soil and vegetation components whereas the two other configurations simulate a composite temperature resulting from the resolution of a composite energy balance with a hot surface layer most of the year.

### 3.3.2 Partition evaporation/transpiration

The plant transpiration measured by the Sapflow method and the stable isotopic technique is compared to those simulated by the three configurations (ISBA-1P, ISBA-2P and ISBA-MEB) during the 2004 and 2013 seasons for the olive and wheat sites, respectively. The simulated transpiration measured by the Sapflow at the Olive orchard site was aggregated at a daily timescale and converted in mm/day. Concerning the isotopic measurements at the wheat site, it was given as the ratio of the total evapotranspiration flux. It is important to state that only the coupled version of ISBA-MEB is able to provide a partition the total evapotranspiration in a bio-physically based maner. Indeed, the single source version of ISBA used in the ISBA-1P and in the ISBA-2P configurations partitions artificially the evapotranspiration based on the cover fraction (Noilhan et Planton, 1989). Table 5 displays the average percentage of transpiration predicted by the three versions and measured by the stable isotope method during the two days of sampling over the wheat site. As expected, the results show that the three configurations give increasing values which are in good accordance with the dynamic of the drying-out of the bare soil. By contrast, values of measured transpiration show an inverse dynamic. This is mainly attributed to the sampling areas which were characterized by a higher percentage of the bare soil for the first day compared to the second one. The problem of the sampling representativeness by the stable isotope method has been detailed in Aouade et al. (2016) using the same data. An average value of the two days was used for comparison in order to improve the observations representativeness. The two configurations show that the transpiration dominates ET and ISBA-1P and ISBA-MEB values are fairly close to the observations in spite of a light underestimation (overestimation) of ISBA-1P (ISBA-MEB).





Figure 7 presents the time series of the plant transpiration simulated by the three configurations and measured by the Sapflow method during the 2004 summer season at the Agdal site. ISBA-MEB outperformed the two other configurations based on the single source version with an RMSE of 0.7 mm/day, a correlation coefficient r=0.73 and a small bias. ISBA-2P predictions are also quite good but with a moderate underestimation of 1 mm/day. By contrast with the dual-sources configurations, the result of the ISBA-1P is significantly worst with an RMSE of about 1.7 mm/day, a low value of r = 0.4

and a strong negative bias of about -1.5 mm/day. Although ISBA-1P and 2-P significantly overestimated the total ET after an irrigation events compared to ISBA-MEB, they underestimated largely the transpiration. This underestimation of ISBA-2P and -1P is in agreement with the higher available energy and, in fine, soil evaporation of these two configurations with regards to ISBA-MEB (cf. time series of predicted soil evaporation as supplementary material; Figure S4). For ISBA-2P, this is because (1) the large soil patch is directly exposed to the incoming solar radiation with no vegetation screening and (2)

there are obviously no roots to extract water on this patch. Indeed, the the evaporation flux for ISBA-2P is of the same order of magnitude as the 1P configuration but because of a strong contrast between the patch "bare soil″ that dominates the total evaporation while the "vegetation" patch evaporation is very low (Figure S4). ISBA-MEB represents also a peak of evaporation after each irrigation event but stills much more moderate than for the other two configurations. This is due to the lower available energy at the ground level than for the other configurations already highlighted. For ISBA-1P, a large part of

this energy is also dissipated by conduction, as was already explained. Finally, the drastic drop of predicted transpiration by ISBA-2P and, to a lesser extent, by ISBA-MEB around mid-august, is probably related to the ability of the olive trees to reach deeper soil layer where water is available. Nevertheless, it is important to keep in mind that the scaling up of Sapflow data from a set of sample trees to the entire plot is a complex processes that relies on an empirical equation. The latter is obtained by plotting the measured total evapotranspiration against the Sapflow observations under dry conditions leading to

lower surface soil moisture and thus the soil evaporation is considered negligible (Er-Raki et al., 2010). This equation is then generalized for wet conditions of Sapflow observations for deriving the stand level pant transpiration. Consequently, this can generate a significant error in estimating stand level plant transpiration as previously reported in several studies (Fernández et al., 2001; Williams et al., 2004; Oishi et al., 2008; Er-Raki et al., 2010).

As a conclusion, although no direct soil evaporation measurements were available, the overall good agreement of ISBA-

535 MEB with transpiration measurements, in particular its small bias, tends to prove that it significantly improves the evapotranspiration partition with regards to the single composite energy budget of ISBA-A-gs in the case of tree cover of moderate sparsity.

## 3.4 Soil hydric budget

A comparison between simulated and observed soil moisture at Agdal and R3 sites is presented in this section. Soil moisture

measurements are available at a half-hourly time steps at the surface layer (5cm) at Agdal site for 2003 season and at 5 and 60cm at the R3 site for the 2003 season.



Figure 8 displays the measured and simulated superficial soil moisture for the wheat and olive sites during the 2003 season and the soil water content in the root zone for the wheat site only. The three configurations show a good agreement with measurements, with moderate RMSE and Bias. However, ISBA-1P tends to dry out the surface layer too fast after each irrigation event, except during the summer month, which support a too high evaporation as already mentioned. This trend to to emphasize evaporation makes the statistical metrics of the single-source configuration slightly worse than the two dual-sources configurations. Interestingly, ISBA-2P and ISBA-MEB provides close prediction of surface soil moisture but, as already highlighted, this is because the high evaporation of the bare soil patch is compensated by the low evaporation of vegetation patch representing a close canopy. During the summer (high evaporative demand), the soil moisture falls to the residual value for ISBA-2P and ISBA-MEB during the severe drought on summer months, due mainly to the deficiency of irrigation between mid-June and August.

## 4 Discussion and conclusion

The present study was carried out in order to evaluate the ability of the multi energy balance version (MEB) of the Interactions Soil-Biosphere-Atmosphere land surface model (ISBA) to simulate the total energy fluxes and its vegetation and soil components including evapotranspiration (ET) and its partition into soil evaporation (E) and plant transpiration ($T_r$) for irrigated crops in semi-arid areas. Two dominating crops of the South Mediterranean region are chosen: an olive orchard and a winter wheat site located in Tensift Al Haouz (Center of Morocco). Observations of ET with Eddy covariance systems, of E and $T_r$ with Sapflow and Isotopic technics were used to validate the performance of ISBA-MEB (coupled scheme) compared to two other configurations of ISBA: 1patch which is the classic big leaf approach (ISBA-1P) and 2 patches which corresponds to a two adjacent component approach (ISBA-2P) or uncoupled scheme.

The contrast of canopy geometries between the two crops leads to significant differences of behavior between the three configurations of the model:

- For an homogeneous cover like wheat, the ability of all the configurations to reproduce the composite soil-vegetation heat fluxes is very close. For the latent heat flux for example, the differences between RMSEs of ISBA-1P, -2P and -MEB do not exceed 10 W/m². These results are consistent with many studies showing that the use of a composite energy balance on homogeneous cover crops is sufficient to provide a good reproduction of convective fluxes (Vogel et al., 1995, Noilhan and Mahfouf 1996). For the olive orchard which represents a quite open canopy (fraction cover of 0.55), both dual-sources configurations outperformed the single-source version.
- A fine analysis of the components of the uncoupled approach (ISBA-2P) shows a strong compensation between fluxes of the bare soil and the vegetation patches for the olive orchard. For instance, evapotranspiration after each irrigation event is strongly overestimated mainly due to strong soil evaporation. This is attributed to a large



available energy at the surface directly exposed to incoming radiation coupled to an absence of root extraction for
the bare soil patch. Stated differently, the aggregated flux is close to the coupled version (ISBA-MEB) and to the
observations but for the "wrong" reasons.

In addition, another specificity of our study focused on irrigated crops in semi-arid areas is the strong transient regime
around an irrigation event leading to a strong shift of energy between sensible and latent heat fluxes. The consequence of the
differences of surface representation between the three model configurations (root distribution, available energy, heat source
coupling …) lead to exacerbated consequences on the energy budget components at this time. Figure 9 summarizes the
behavior of the three configurations around an irrigation event. It displays the average time series of predicted and observed
surface temperature ($T_s$), ground heat flux (G), and the convective heat fluxes (H and LE) from 5 days before to 8 days after
an irrigation event. Irrigation causes obviously a drop of the composite soil/vegetation temperature (Figure 9a). The energy
is therefore mainly attributed to the latent heat flux (Figure d) at the expense of the sensible heat flux (Figure 9c). This is
predicted by the three configurations of the model but with different level of accuracy. The differences of the configuration
behaviors at this time explain, to a large extent, the differences in the overall performance between the simple balance
configuration and the two others. ISBA-1P shows abnormally high values of LE after an irrigation event. By contrast, ISBA-
2P and above all, ISBA-MEB are able to better reproduce the observed moderate shifting.

One of the main conclusions of the study is that the new ISBA-MEB version implemented in the SURFEX platform has
proved to be more suitable than single source configuration for estimating convective fluxes including evapotranspiration
and its components, at least for moderately open tree canopies. This shows the need to take into account the interaction
between vegetation and soil acting as coupled sources of heat in the parameterization of SVATs when vegetation is sparse.
The choice between coupled and uncoupled model, to better represent exchanges between the biosphere and the atmosphere,
is not straightforward anyway. The obtained results demonstrated that the coupled energy balance provided also the best
estimates of components and composite fluxes but the patch approach followed closely. Likewise, this study also showed, as
suggested by Choudhury and Monteith (1988), that the new parameters introduced in ISBA-MEB (such as the attenuation
coefficient for momentum and for wind) are highly sensitive and vegetation-type dependent as evidenced by the different
calibrated values between the two studied crops. This study points out the need for further local scale evaluation on different
crops of various geometry (more open rainfed or denser intensive olive orchard) and over different climatic conditions in
order to assess in particular from which degree of sparsity, a dual source approach should be preferred. This will both further
our understanding of the representation of soil and vegetation heat sources in the SURFEX platform and also help to provide
adequate parameterization to global data base such as ECOCLIMAP-II in the view of a global application of the ISBA-MEB
model. Finally, considering the heavy trend towards the conversion of traditional (wheat) crops to tree crops in the south
Mediterranean region, which are more financially attractive but that also consume more water (Jarlan et al., 2016),

improving the representation of complex crops in SVAT model is also of prime importance for future studies on surface-atmosphere retro-action or global change impact.

**Code and data availability**

The MEB code is a part of the ISBA LSM and is available as open source via the surface modeling platform called
SURFEX, which can be downloaded at http://www.cnrm-game-meteo.fr/surfex/. Validation data on both sites may be distributed on request to the co-leads of the Tensift observatory Pr. Jamal Ezzahar (j.ezzahar@uca.ma) and Dr. Vincent Simonneaux (vincent.simonneaux@ird.fr)

**Acknowledgements**

This work has been carried out within the frame of the Joint International TREMA (IRD, UCAM, DMN, CNESTEN,
ABHT, ORMVAH) and of the ERANETMED03-62 CHAAMS project. Financial supports for the experiment have been provided by IRD, the MISTRALS/SICMED program through the METASIM project, the ANR AMETHYST project (ANR-12-TMED-0006-01), the SAGESSE project (Projet Prioritaire de Recherche PPR - Type B), funded by the Minister for Higher Education, Scientific Research and Executive Training (Morocco), PHC Toubkal TBK/18/61 and the RISE REC project (H2020/645642) financed by the Marie Skłodowska-Curie Research and Innovation Staff Exchange (RISE). G.
Aouade received a travel grant from the MISTRALS/ENVIMED program through the CHAMO project and a financial support from MISTRALS/SICMED2.

**Author contributions**

Aouade G., Jarlan L., worked out almost all of the technical details, performed the numerical calculations for the suggested experiment and took the lead in writing the manuscript. Jarlan L., Ezzahar J. and Boone A., conceived the original idea and
supervised the findings of this work. Er-raki S., Benkaddour A., Khabba  S., Boulet G., Garrigues S., Chehbouni A., Boone A.,  Napoly A., contributed to the interpretation of the results and to the final version of the manuscript. Boone A., Napoly A., designed the model and the computational framework. All authors provided critical feedback and helped shape the research, analysis and manuscript.

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



## Tables

**Table 1**: Input parameter and variables of the ISBA model derived from *in situ* measurements.

| | Olive orchard site | Wheat site |
|---|---|---|
| **Patch** | Temperate Broad Leaf Evergreen | Crop C3 |
| **Cover fraction (%)** | 55 | Variable |
| **LAI (m²/m²)** | 3 | Variable |
| **Vegetation height (m)** | 6.5 | Variable |
| **Emissivity** | 0.98 | 0.97 |
| **Soil albedo** | 0.18 | 0.15 |
| **Vegetation albedo** | 0.14 | 0.20 |
| **Soil texture (%)** | 44% Sand, 30% Clay | 20% Sand, 47% Clay |
| **Root depth (m)** | 1 | 0.55 |


**Table 2**: List of parameters used for sensitivity analysis and their considered ranges.

| Name | Description | Parameter range | Unit | References |
|---|---|---|---|---|
| Sd | Sand content | 0.39-0.48 (Agdal) 0.18-0.22 (R3) | - | Noilhan and Mahfouf, 1995; Equation (27) |
| Cl | Clay content | 0.27-0.33 (Agdal) 0.42-0.51 (R3) | - | Noilhan and Mahfouf, 1995; Equation (28) |
| RD | Root depth | 0.5-1 (Perennial trees) 0.4-0.7 (C3 crops) | m | ECOCLIMAP |
| $Z_0/Z_{0H}$ | Roughness ratio | 9-11 (Agdal) 7-11 (R3) | - | Boone et al., 2017; Equation (66) |
| $\tau_{LW}$ | Longwave radiation transmission factor | 0.1-0.9 | - | Boone et al., 2017; Equation (45) |
| $g_m$ | Mesophyll conductance | 0.0001- 0.04 | m.s$^{-1}$ | Calvet 2000 ; Equation (A1) |
| $\Gamma$ | Coefficient for the calculation of the surface stomatal resistance | 0-0.06 | µmol mol$^{-1}$ | Calvet 2000 ; Equation (A1) |



| | | | | |
|---|---|---|---|---|
| $W_{r\max}$ | Coefficient for maximum water interception storage on capacity on the vegetation | 0.05-0.3 | mm | Noilhan and planton 1989; Equation (24) |
| $C_v$ | Thermal coefficient for the vegetation canopy | 0.5e-5 - 3e-5 | J.m².K$^{-1}$ | Noilhan and planton 1989; Equation (8) |
| $g_c$ | Cuticular conductance | 0-0.0004 | m.s$^{-1}$ | Gibelin et al., 2006; Equation (A3) |
| $U_l$ | Typical value of wind speed | 0.5- 3 | m.s$^{-1}$ | Sellers et al., 1996; Equation (B7) |
| $\theta_c$ | Critical normalized soil water content for stress parameterization | 0.1-0.5 | – | Calvet 2000 ; Equation (9) |
| $D_{\max}$ | Maximum air saturation deficit | 0.03-0.6 | kg.kg$^{-1}$ | Calvet 2000; Equation (A3) |
| $L_w$ | Leaf width | 0.01-0.04 | m | Boone et al., 2017; Equation (51) |
| $\phi_v$ | Attenuation coefficient for momentum | 1.5-5 | – | Boone et al., 2017; Equation (55) |
| $\phi'_v$ | Attenuation coefficient for wind | 2- 4 | – | Boone et al., 2017; Equation (51) |

**Table 3**: Default values from literature or ECOCLIMAP-II data base and optimal values (see text) of the highly sensitive parameters for one of the objective functions.

| Olive/Agdal | Default values | Optimal values | Wheat/R3 | Default values | Optimal values |
|---|---|---|---|---|---|
| Sand | 0.44 | 0.47 | Clay | 0.47 | 0.44 |
| Clay | 0.30 | 0.27 | RD | 0.50 | 0.55 |
| RD | 1.00 | 0.62 | $\frac{z_0}{z_{0h}}$ | 10.00 | 7.01 |
| $\tau_{LW}$ | 0.50 | 0.43 | $\tau_{LW}$ | 0.40 | 0.31 |
| $U_l$ | 1.000 | 2.435 | $U_l$ | 1.00 | 1.96 |





| $L_w$ | 0.010 | 0.021 | $L_w$ | 0.010 | 0.029 |
|---|---|---|---|---|---|
| $\phi_v$ | 2.00 | 4.45 | $\phi_v$ | 2.00 | 3.12 |
| $\phi_v'$ | 3.00 | 2.12 | $\phi_v'$ | 3.00 | 2.24 |

**Table 4:** Comparison between observations and ISBA for the components of the energy balance through Root mean square error (RMSE), correlation coefficient (r) and bias (BIAS). The calibration period is 2003 for both sites while validation is 2004 for Agdal and 2013 for R3.

| | | | Calibration | | | Validation | | |
|---|---|---|---|---|---|---|---|---|
| | | | RMSE | r | BIAS | RMSE | r | BIAS |
| ISBA-1P | Wheat | $R_n$ | 27.8 | 0.98 | 0.9 | 24.8 | 0.99 | -5.1 |
| | | G | 19.7 | 0.83 | 1.0 | 32.8 | 0.68 | 21.9 |
| | | H | 55.2 | 0.71 | 4.3 | 31.7 | 0.62 | -10.1 |
| | | LE | 82.7 | 0.73 | 1.9 | 69.2 | 0.85 | 15.8 |
| | Olive orchard | $R_n$ | 19.0 | 0.99 | -3.0 | 14.8 | 0.99 | 0.9 |
| | | G | 22.7 | 0.73 | 11.9 | 42.0 | 0.75 | 37.0 |
| | | H | 76.0 | 0.68 | -9.0 | 90.3 | 0.60 | -41.9 |
| | | LE | 86.1 | 0.52 | 4.0 | 107.1 | 0.67 | 31.5 |
| ISBA-2P | Olive orchard | $R_n$ | 17.2 | 0.99 | -3.0 | 11.0 | 0.99 | -0.7 |
| | | G | 28.2 | 0.68 | -7.5 | 17.0 | 0.78 | 14.6 |
| | | H | 40.4 | 0.85 | 11.6 | 47.0 | 0.83 | -12.0 |
| | | LE | 46.3 | 0.81 | 9.4 | 52.2 | 0.86 | 12.7 |
| ISBA-MEB | Wheat | $R_n$ | 29.0 | 0.98 | -2.8 | 26.0 | 0.99 | -5.5 |
| | | G | 32.2 | 0.65 | 17.9 | 55.2 | 0.79 | 31.7 |
| | | H | 55.0 | 0.66 | -10.8 | 26.2 | 0.73 | 0.5 |
| | | LE | 73.7 | 0.73 | -0.9 | 56.7 | 0.92 | -8.0 |
| | Olive orchard | $R_n$ | 17.7 | 0.99 | -3.0 | 11.0 | 1.0 | -1.9 |
| | | G | 29.0 | 0.69 | 21.8 | 54.0 | 0.74 | 49.0 |
| | | H | 39.0 | 0.85 | -0.6 | 44.0 | 0.84 | -15.0 |
| | | LE | 38.6 | 0.83 | -4.9 | 40.2 | 0.88 | -3.4 |






**Table 5 :** Percentages of transpiration simulated by the three configurations of the ISBA model and measured by the isotopic method for the 2013 season at the R3 site. The corresponding LAI and $F_c$ are also provided.

|  | LAI(m²/m²) | $F_c$ | $T_r$ (ISBA-1P) | $T_r$ (ISBA-2P) | $T_r$(ISBA_MEB) | $T_r$ (Observations) |
|---|---|---|---|---|---|---|
| **101** | 3.8 | 0.9 | 73% | 60% | 85% | 83% |
| **102** | 3.8 | 0.9 | 75% | 61% | 89% | 77% |

**Figures captions**

Figure 1: Overview of the two experimental sites: Olive orchard named Agdal and winter wheat named R3.

Figure 2: Schematic description of the three configurations of ISBA model: (a) the single-source configuration (ISBA-1P); (b) the layer configuration (ISBA-MEB); (c) the patch configuration (ISBA-2P).

Figure 3: results of the sensitivity analysis of the ISBA-MEB model for both sites.

Figure 4: Time series of the simulated and measured latent heat flux (LE) for the Agdal site (2003 and 2004 seasons) and for the R3 site (2003 and 2013 seasons).

Figure 5: Taylor diagrams for the net radiation $R_n$, the sensible heat flux H and the ground heat flux for Agdal (2003 and 2004 seasons) and R3 sites (2003 and 2013 seasons). ISBA-1P is in red, ISBA-2P is in blue, ISBA-MEB is in green and observations are indicated using a black point.

Figure 6: Time series of the soil net radiation (a; only the patch ''bare soil'' is shown for the ISBA-2P configuration), soil temperature (b) and vegetation temperature (c) simulated by the three configurations and measured at the Agdal site for the 995    2003 season.

Figure 7: Time series of the simulated plant transpiration and measurements by the Sapflow method during the 2004 summer season at the Agdal site.

Figure 8: Comparison between the simulated soil water content with that measured at 5cm at the R3 and Agdal sites during the 2003 season, as well as for the root zone (60cm for the R3 site only).

Figure 9: Time series of simulated and observed surface temperature ($T_s$), the ground heat flux (G), and the convective heat fluxes (H and LE) during the transient regime around an irrigation event (from 5 days before irrigation and to 8 days after irrigation).






**Figures**

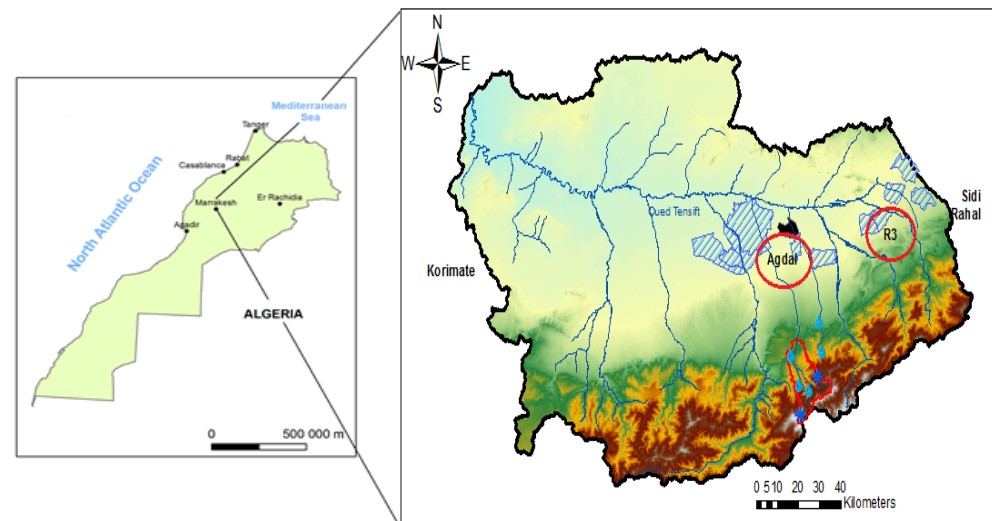

**Figure 1**

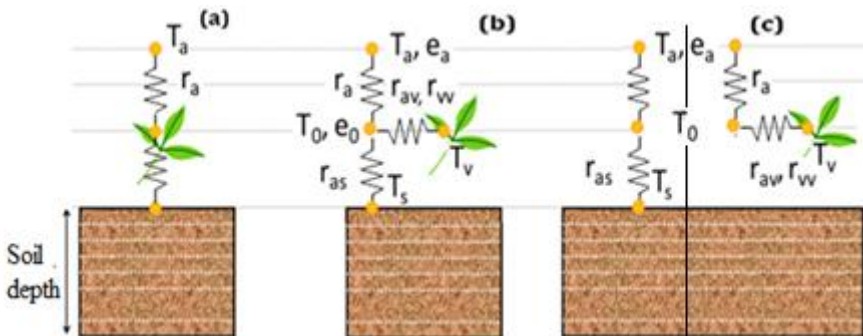

**Figure 2**





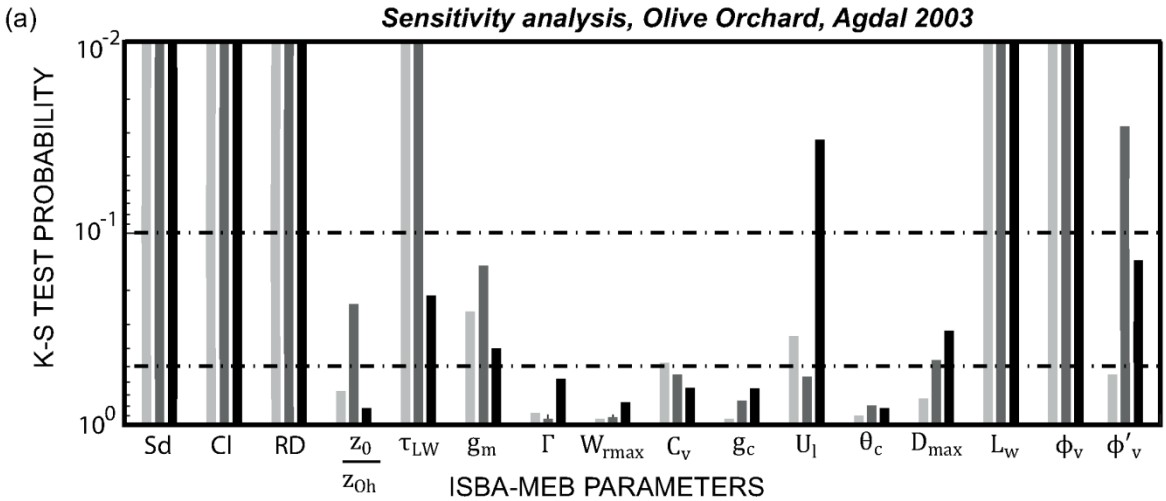

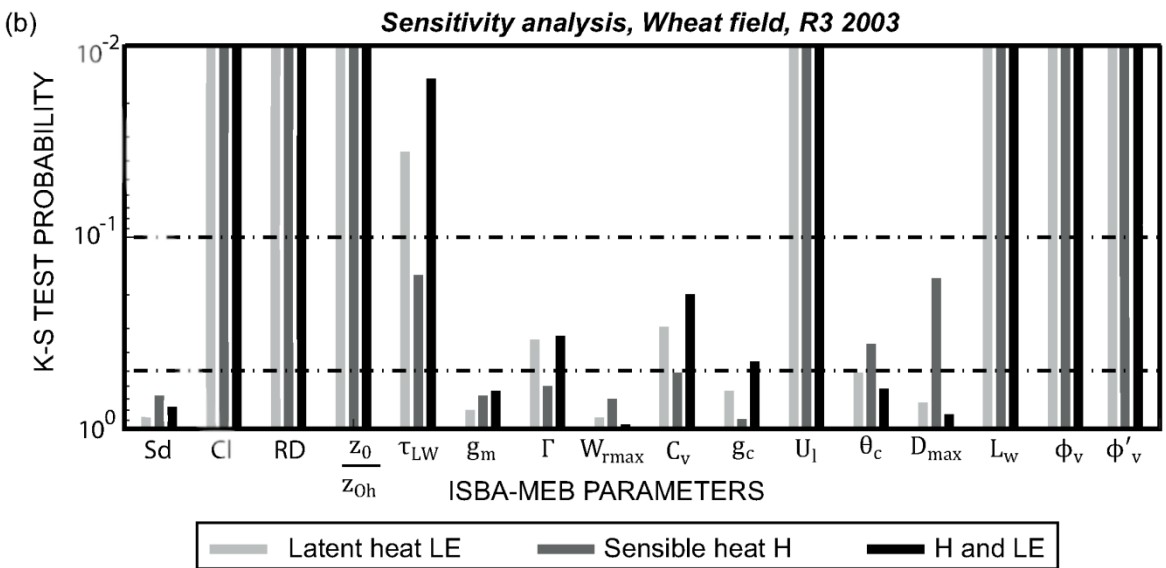

**Figure 3**







**Figure 4**



**Figure 5**





**Figure 6**

1030


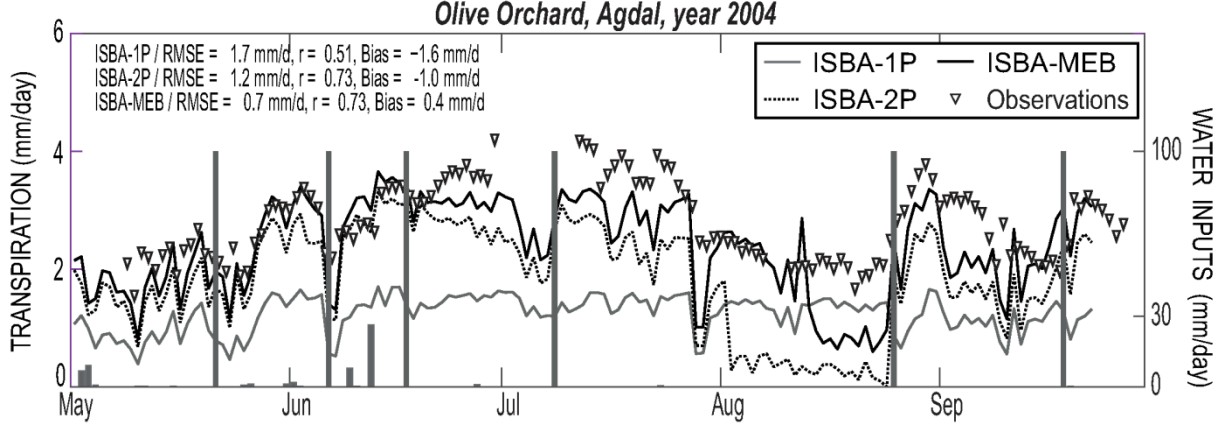

**Figure 7**



**Figure 8**

1035


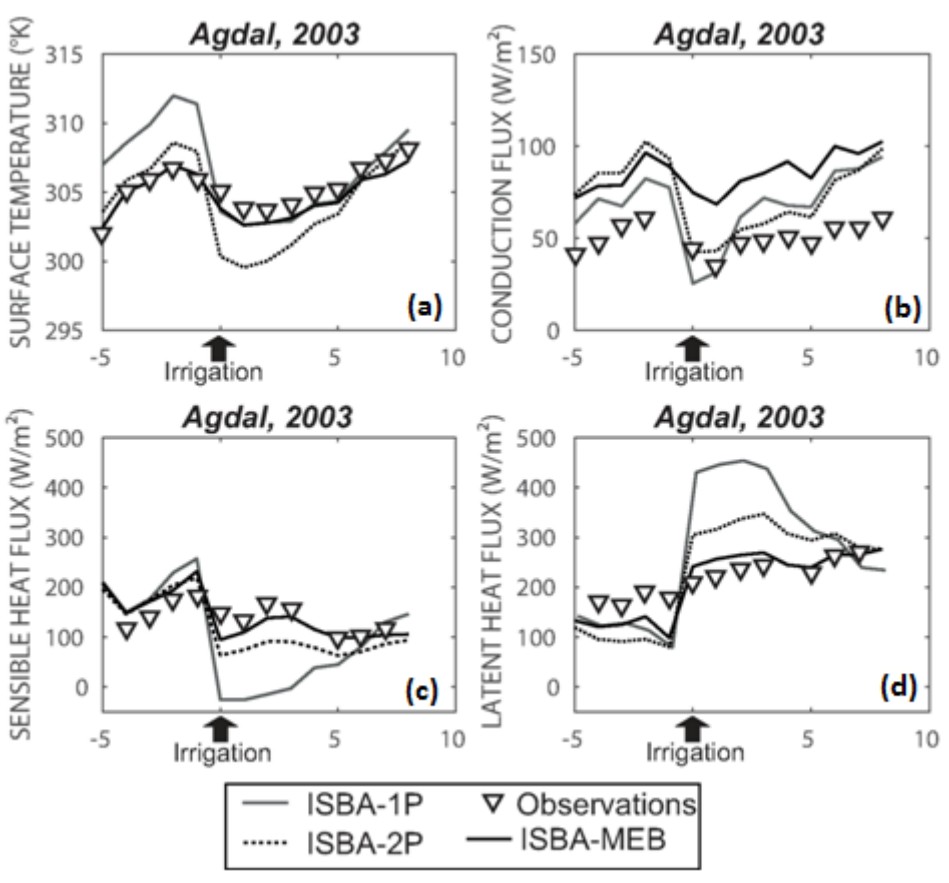

**Figure 9**

## Appendix 1: Prognostic Equations of ISBA-A-gs

Standard version

The governing equations for heat and water transfers within the soil and at the surface are given by the following formulas (More details are provided in Decharme et al., 2011):

$$\frac{\partial T_{g,1}}{\partial t} = C_T G_0 - C_G \frac{\overline{\lambda_1}}{\Delta \tilde{z}_1} \left( T_{g,1} - T_{g,2} \right) \tag{1}$$

$$\frac{\partial T_{g,i}}{\partial t} = \frac{1}{C_{g,i}\Delta_{g,i}} \left[ \frac{\overline{\lambda_{i-1}}}{\Delta \tilde{z}_{i-1}} \left( T_{i-1} - T_i \right) - \frac{\overline{\lambda_i}}{\Delta \tilde{z}_i} \left( T_i - T_{i+1} \right) \right] \forall i = 2, N \tag{2}$$





$$\frac{\partial W_{g,1}}{\partial t} = \frac{1}{\rho_w \Delta z_{g,1}} \left[ (1 - veg)P_r + D_r - E_g - R - F_{g,1} \right] \tag{3}$$

$$\frac{\partial W_{g,i}}{\partial t} = \frac{1}{\rho_w \Delta z_{g,i}} \left( F_{g,i} - F_{g,i+1} \right) \tag{4}$$

Where $T_{g,1}$ (K) is the uppermost ground temperature and $T_{g,i}$ (K) is the temperature of the layer i; $C_T$ (K.m$^{-2}$.J$^{-1}$) is the surface composite thermal inertia coefficient (Noilhan and Planton, 1989); $G_0$ is the flux between the atmosphere and the surface; $\Delta\tilde{z}_i$ (m) and $\Delta z_i$ (m) are the thickness between two consecutive layer midpoints or nodes and the thickness of the layer i, respectively; $C_{g,i}$ (J.m$^{-3}$.K$^{-1}$) is the heat capacity of the soil, $\overline{\lambda_i}$ (W.m$^{-1}$.K$^{-1}$) is the inverse of weighted arithmetic mean of the soil thermal conductivity at the interface between two consecutive nodes and veg is the cover fraction. $F_{g,i}$ represents the vertical flow of water between layers i and i+1 and is given by the Law of Darcy. $E_g$, $D_r$, $P_r$ and R are the amount of water evaporated from the soil (Kg.m$^{-2}$.s$^{-1}$), rainfall (Kg.m$^{-2}$.s$^{-1}$), canopy drainage (Kg.m$^{-2}$.s$^{-1}$) and surface runoff (Kg.m$^{-2}$.s$^{-1}$).

The soil heat flux, G, is defined as:

$$G = \frac{\overline{\lambda_i}}{\Delta\tilde{z}_i} \left( T_i - T_{i+1} \right) \tag{5}$$

With $\overline{\lambda_i}$ is the thermal conductivity, $\Delta\tilde{z}_i$ the thickness between the center of the first layer and that of the second, $T_i$ the temperature of the first layer and $T_{i+1}$ the temperature of the second layer.

The net radiation is calculated as follows:

$$R_n = R_G(1 - \alpha) + \varepsilon(R_A - \sigma T_s^4) \tag{6}$$

Where $\alpha$ and $\varepsilon$ are the albedo and the emissivity, respectively, $\sigma$ (W.m$^{-2}$.K$^{-4}$) is the Stefan Boltzmann constant, $R_G$ is the incoming solar radiation and $R_A$ is the atmospheric radiation.

The sensible heat flux, H, is expressed as follows:

$$H = \rho_a c_p C_H V_a (T_s - T_a) \tag{7}$$



Where $\rho_a$ (Kg.m$^{-3}$) is the air density, $c_p$ (J.Kg$^{-1}$.K$^{-1}$) is the specific heat of the air, $V_a$ (m.s$^{-1}$) is the wind speed and $T_a$(k$^{-1}$) is

air temperature, $C_H$ is the drag coefficient.

The latent heat flux $LE$ (W.m$^{-2}$), the evaporation from the soil ( $E_g$ ), the direct evaporation from the foliage ( $E_r$ ) and the

transpiration ( $E_{tr}$ ) are defined as follows:

$$LE = L_v\left(E_r + E_{tr} + E_g\right) \tag{8}$$

$$E_g = (1 - veg)\rho_a C_H V_a[h_u q_{sat}(T_s) - q_a] \tag{9}$$

$$E_r = veg\frac{\delta}{R_a}\rho_a C_H V_a[q_{sat}(T_s) - q_a] \tag{10}$$

$$E_{tr} = veg\frac{1-\delta}{R_a+R_s}\rho_a C_H V_a[q_{sat}(T_s) - q_a] \tag{11}$$

Where $L_v$ (J.kg$^{-1}$) is the latent heat of vaporization, , $q_{sat}(T_s)$ (kg.kg$^{-1}$) is the saturated specific humidity at the surface

temperature $T_s$ and $q_a$ (kg.kg$^{-1}$) is the atmospheric specific humidity at the lowest atmospheric level. $h_u$ is the relative

humidity at the ground surface. $\delta$ is the vegetation fraction that covered by intercepted water. $R_a$ and $R_s$ (s.m$^{-1}$) are the
aerodynamic and canopy surface resistances, respectively.

Additionally, in this study, ISBA uses the A-g$_s$ parameterization to estimate the stomatal conductance $g_s$, by considering the

impact of the atmospheric carbon dioxide concentration and the interactions between all environmental factors on the

stomatal aperture. Therefore, the leaf stomatal conductance is expressed as follow (Calvet et al., 1998):

$$g_s = g_c + 1.6\left(A_n - A_{min}\left(\frac{D_s}{D_{max}^*}\frac{A_n+R_d}{A_m+R_d}\right) + R_d\left(1 - \frac{A_n+R_d}{A_m+R_d}\right)\right)/(C_s - C_i) \tag{12}$$

Where $g_c$ (mm.s$^{-1}$) is the cuticular conductance, $A_n$ (mg.m$^{-2}$.s$^{-1}$) is the net assimilation, $A_{min}$ is the rate of the residual

photosynthesis rate (at full light intensity). C$_s$ and C$_i$ are the internal and air CO$_2$ concentrations, respectively. $D_s$ and $D_{max}^*$

are the leaf-to-air saturation deficit and $D_{max}^*$ the maximum leaf-to-air saturation deficit, respectively. $A_m$ is the

photosynthesis rate in light-saturating conditions.


The $g_s$ is multiplied by the Leaf Area Index (LAI) value in order to scale up $g_s$ from the leaf to the canopy. Finally, the integrated canopy net assimilation $A_{nI}$ and conductance $g_{sI}$ which were used to compute the heat and water vapor surface fluxes and the canopy resistance, respectively, are then written by assuming an homogeneous leaf vertical distribution as follows:

$$A_{nI} = LAI \int_0^1 A_n d(z/h) \tag{13}$$

$$g_{sI} = LAI \int_0^1 g_s d(z/h) \tag{14}$$

With $h$ is the height of the canopy and $z$ is the distance to the soil.

ISBA Multi-Energy Balance (MEB)

Compared to ISBA standard, MEB distinguishes the soil and the vegetation surface temperatures. Fluxes from the ground or vegetation first transit to the so called "canopy air space" or "canopy" before being in contact with the atmosphere. For an extended details about the prognostic equations and its numerical resolution aspects as well as the various assumptions of the MEB version, the reader can refer to Boone et al. (2017). We develop in the following paragraphs the main equations and parameterizations that will be used for this study. As soil freeze thaw is negligible for this study (no snow process involved),

the related terms will be not presented.

The prognostic equations for the energy budget are:

$$C_v \frac{\partial T_v}{\partial t} = R_{nv} - H_v - LE_v \tag{15}$$

$$C_{g,1} \frac{\partial T_{g,1}}{\partial t} = R_{ng} - H_g - LE_g - G_{g,1} \tag{16}$$

Where $T_{g,1}$ is the uppermost ground temperature, and $T_v$ is the bulk canopy temperature (K). The subscripts 1, $v$ and $g$

indicate the uppermost layer, vegetation canopy and ground. The ISBA-MEB sensible heat fluxes, $H_v$ (between the vegetation and canopy air space), $H_g$ (between the ground and canopy) and $H_c$ (between the canopy and the overlaying atmosphere) are defined as:

$$H_v = \rho_a \frac{(T_v - T_c)}{R_{av-c}} \tag{17}$$


$$H_g = \rho_a \frac{(\Gamma_g - \Gamma_c)}{R_{ag-c}} \tag{18}$$

$$H_c = \rho_a \frac{(\Gamma_c - \Gamma_a)}{R_{ac-c}} \tag{19}$$

$R_{ag-c}$, $R_{av-c}$ and $R_{ac-a}$ are s the aerodynamic resistances to heat transfer between: the canopy and the ground (Choudhury and Monteith, 1988), the canopy and the vegetation and the atmosphere and the canopy, respectively. $\Gamma$ (J.kg$^{-1}$) is a thermodynamic variable which is expressed as a linear relationship with the temperature (Boone et al., 2017). Note that in the model code, potential temperature or dry static energy are used as thermodynamic variables, but for simplicity, these

quantities have been approximated using temperature (since the impact of this approximation is quite small in the current study).

The ISBA-MEB latent heat fluxes, $E_v$ (between the vegetation and canopy), $E_g$ (between the ground and canopy) and $E_c$ (between the canopy and the overlaying atmosphere) are defined as:

$$E_v = \rho_a h_{sv} \frac{(q_{satv} - q_c)}{R_{av-c}} \tag{20}$$

$$E_g = \rho_a \frac{(q_g - q_c)}{R_{ag-c}} \tag{21}$$

$$E_c = \rho_a \frac{(q_c - q_a)}{R_{ac-a}} \tag{22}$$

Where $h_{sv}$ is the Halstead coefficient.

In what follows, we present the main parameterizations introduced to calculate the new parameters needed by the model, as the fraction of the vegetation covered with intercepted water:

$$\delta_v = (1 - \omega_{rv}) \left( \frac{W_r}{W_{r\max}} \right)^{2/3} + \frac{\omega_{rv} W_r}{(1 + \alpha_{rv} LAI) W_{r\max - \alpha_{rv}} W_r} \tag{23}$$

The maximum water interception storage on capacity on the vegetation is defined simply as:

$$W_{r\max} = c_{wrv} LAI \tag{24}$$

With $c_{wrv} = 0.2$

The canopy absorption is defined as:





$\sigma_{LW} = 1 - \exp(-\tau_{LW} LAI)$ (27)

Where $\tau_{LW}$ represents a longwave radiation transmission factor that can be species (or land classification) dependent.

The aerodynamic resistance between the vegetation canopy and the surrounding air space is defined as:

$R_{avg-c} = \left(g_{av} + g_{av}^*\right)^{-1}$ (28)

The bulk canopy aerodynamic conductance $g_{av}$ between the canopy and the canopy air is parameterized as follows
(Choudhury and Monteith, 1988):

$g_{av} = \dfrac{2 LAI \alpha_{av}}{\phi_v'} \left(\dfrac{u_{hv}}{l_w}\right)^{1/2} \left[1 - \exp\left(-\phi_v'/2\right)\right]$ (29)

Where $u_{hv}$ is the wind speed at the top of the canopy (m.s$^{-1}$), $l_w$ is the leaf width, $\alpha_{av}$ is the canopy conductance scale factor, $\phi'_v$ is the attenuation coefficient for wind.
