# Peer review of "Evapotranspiration partition using the multiple energy balance version of the ISBA-A-gs land surface model over two irrigated crops in a semi-arid Mediterranean region (Marrakech, Morocco)"

_Hydrology and Earth System Sciences, 2019_

## Referee Comment (RC1) · Anonymous Referee #1 · 19 Nov 2019

This paper compares the behavior of several energy balance modelling schemes over two contrasting crops in Morocco. The study cases are well selected, as both represent major crops in the Mediterranean basin, with very different canopy structure and growth dynamics. Such comparisons, single vs. double-source approaches, or uncoupled vs. coupled double source models have been performed since the development of these models in the 90s. Therefore, even if the general approach is not new, it contributes to accumulate experience in energy balance modelling applications over different landcovers and climatic conditions. In addition, it can be of interest for users of the ISBA model and the recent multiple balance version (ISBA-MED) and it also provides some insight about the partition of the latent heat. However, the following aspects of the paper require further clarifications or discussion:

- The information provided about the meteorological, micrometeorological and soil validation data is insufficient to evaluate their quality and support the main results and conclusions of the study (the only reference of this section, Hoedjes et al. (2007), is not provided in the reference list). References to other papers, included in the descriptions of the sites, might be useful to access such information, but the minimum data necessary to evaluate the work should be included in the paper. In particular: o It lacks general information about the quality of EB measurements at both sites (eg. closure values obtained during the different measurement periods). o Soil net radiation observations are presented but it is not explained how it was measured. o The same thing happened with the soil/vegetation temperatures, and the surface temperature. The latter variable is a primary boundary condition to estimate energy balance components, and it should be mentioned how it was obtained and how the separation into vegetation and soil temperatures was performed, which is a difficult task and one of the main limitations for the applicability of two-layer representations. - Another difficulty for interpreting the results is the confusion of lines, with line types sometimes difficult to differentiate and markers (eg. Figure 4, 6, 8) creating linear features easy to be confused with real lines. - A direct interpretation of the results indicates that ISBA-MED outperforms all other versions of the model for both canopies. However, the authors interpret that this is only clear for olive trees and that for wheat the 1P and MED versions perform similarly. It is striking that ISBA-MED accuracy is better for a discontinuous and more heterogeneous tree crop as an olive grove than for homogeneous wheat, also better than 1P for wheat. Do the authors have a plausible explanation? - It is concluded that 1-P accuracy is "sufficient" for the wheat because both models perform similarly, but a measurement of percentage error or average LE values is not provided. Without this information, it is not possible to get an idea of the real utility of these estimations. -

According to water inputs observed in 4, the wheat was barely irrigated during the first season. It should have suffered severe water stress, with a poor development. Could this issue have affected the calibration of the different models over wheat? It would be useful to interpret the results to add a brief description of crop conditions during the different seasons in the site description. - Were olive groves maintained free of grasses all year round? The appearance of a grass layer between olive trees during part of the growing season is quite often. It could be an intended management practice or occur naturally and not be properly removed. Either way, it would highly affect the balances of energy and water, and it should be mentioned. - A few sentences of the abstract are unclear (lines 31-40), with confusing and sometimes erroneous references to the different versions and crops. For example, in lines 31-33, it should be specified, within the sentence, that it makes reference only to the results on wheat, and 2P is not applied on wheat. On the next sentence, starting "By contrast" it is not clear if the contrast is because of the crop or due to the model, as 2P is only applied to olive trees. On the next "By contrast" (line 37), it is not clear to which contrast the authors are referring. - Line 102. Kustas and Norman, 1997 or 1996? Please correct the reference if it is really useful. The paper presents a review of many models not specifying a patch representation. The other two references: Norman et al. 1995 and Boulet et al. 2015 make reference to both, parallel and series, schemes. - Please check the wheat site coordinates. It probably should be 31°38' instead of 31°68'. - Line 187: What "Similarly to R3 site" means here? - Line 362. Did daily calculations include nighttime? - Conclusions. Line 567. -2P was not applied to wheat.

---

## Referee Comment (RC2) · Pierre Gentine (Referee) · 13 Feb 2020

This paper by Aouade et al demonstrates the potential of different complexity of the multi-source approaches to surface flux partitioning. The paper is quite clear and correctly organized. The strategy and methodology are sound. The conclusions are supported by the results. My comments are really minor, mostly related to some editing of the text. The authors did a good job in this submission, I believe. My detailed

comments are attached in a pdf.

Please also note the supplement to this comment:
https://www.hydrol-earth-syst-sci-discuss.net/hess-2019-532/hess-2019-532-RC2-supplement.pdf

[revised manuscript text omitted]

**Figures**

[Figure]

**Figure 1**

[Figure]

**Figure 2**

[Figure]

[Figure]

[Figure]

[Figure]

**Figure 3**

[Figure]

**Figure 4**

[Figure]

**Figure 5**

[Figure]

(a)

**Olive Orchard, Agdal, year 2003**

ISBA-1P / RMSE = 101.3 W/m², r = 0.92, Bias = 55.7 W/m²
ISBA-2P / RMSE = 96.8 W/m², r = 0.91, Bias = 27.7 W/m²
ISBA-MEB / RMSE = 240.8 W/m², r = 0.90, Bias = -193.5 W/m²

(b)

**Olive Orchard, Agdal, year 2003**

ISBA-1P / RMSE = 6.4 °K, r = 0.93, Bias = -1.1 °
ISBA-2P / RMSE = 6.2 °K, r = 0.93, Bias = 2.3 °
ISBA-MEB / RMSE = 5.9 °K, r = 0.96, Bias = -1.7 °

(c)

**Olive Orchard, Agdal, year 2003**

ISBA-1P / RMSE = 4.5 °K, r = 0.93, Bias = 2.7 °
ISBA-2P / RMSE = 4.6 °K, r = 0.95, Bias = 3.2 °
ISBA-MEB / RMSE = 4.2 °K, r = 0.95, Bias = 2.4 °

1030

**Figure 6**

[Figure]

[Figure]

[Figure]

**Figure 7**

[Figure]

**Figure 8**

[revised manuscript text omitted]

---

## Author Comment (AC1) · 26 Mar 2020

We wish to thank the reviewer for his interest and his meaningful comments. All of them have been addressed. We hope that the paper has been improved. See detailed responses below.

1.1. The information provided about the meteorological, micrometeorological and soil validation data is insufficient to evaluate their quality and support the main results and

conclusions of the study. (the only reference of this section, Hoedjes et al. (2007), is not provided in the reference list). References to other papers, included in the descriptions of the sites, might be useful to access such information, but the minimum data necessary to evaluate the work should be included in the paper.

Agree. According to the reviewer's suggestion, the section was rewritten (see Lines 145-189) and a new table summarizing the different instruments used over both sites was added in order to provide more information on the meteorological, micrometeorological and soil validation data. In addition to Hoedjes et al. (2007 and 2008), the references to Ezzahar et al. (2007a, 2007b, 2009a, 2009b) and to Duchemin et al (2008), who used the same data sets were also added.

1.2. It lacks general information about the quality of EB measurements at both sites (eg. closure values obtained during the different measurement periods).

Agree. According to the reviewer comment, a new paragraph providing information about the quality of the energy balance during the different measurement periods over both sites was added (please see lines 191-197).

1.3. Soil net radiation observations are presented but it is not explained how it was measured.

Agree. The soil net radiation over the olive tree orchard was measured over bare soil at 1m height using a Q7 radiometer. According to the reviewer's suggestion, this is now detailed in the new version of the manuscript (see lines 170-172). In addition, a table was added which contains the different micro-meteorological instruments (see point 1.1 above).

1.4. The same thing happened with the soil/vegetation temperatures, and the surface temperature. The latter variable is a primary boundary condition to estimate energy balance components, and it should be mentioned how it was obtained and how the separation into vegetation and soil temperatures was performed, which is a difficult
task and one of the main limitations for the applicability of two-layer representations.

Agree. Over the Agdal site, which is an open orchard, soil/vegetation temperatures were measured separately using two infrared thermometers (IRTS-Ps), with a 3:1 field of view, at heights of 1m (pointing towards the soil) and 7.15 m (pointing towards the crown of the tree), respectively. The surface temperature was derived from the CNR1 radiometer which was placed at 8.5 m height in order to embrace vegetation and soil radiances by ensuring that the field of view was representative of their respective cover fractions. On the contrary, only one IRTS-Ps installed at 2m was used to measure the composite surface temperature over wheat site. According to the reviewer comments, this is now detailed in the new version of the manuscript (See lines 172-175).

1.5. The confusion of lines, with line types sometimes difficult to differentiate and markers (eg. Figure 4, 6, 8) creating linear features easy to be confused with real lines.

OK. We initially plotted the figures with colors and change them for black and white versions before submission. According to the reviewer's suggestion, the figure 4, 6 and 8 (and figure S2) have been redrawn with colors instead of different lines type. We agree that lines are easier to differentiate.

1.6. A direct interpretation of the results indicates that ISBA-MEB outperforms all other versions of the model for both canopies. However, the authors interpret that this is only clear for olive trees and that for wheat the 1P and MEB versions perform similarly. It is striking that ISBA-MEB accuracy is better for a discontinuous and more heterogeneous tree crop as an olive grove than for homogeneous wheat, also better than 1P for wheat. Do the authors have a plausible explanation ?

OK. For wheat, it is clear from the statistical metrics and the time series that ISBA-MEB and ISBA-1P are very close: on LE, RMSE differences between both versions are lower than 10W/m$^2$ with same correlation coefficients and very low biases for the calibration year (see table 4). Performances are also close on the validation year. By contrast, the

differences between ISBA-MEB and ISBA-1P are much higher for the Olive orchard for both calibration and validation years. Generally speaking, the difference between ISBA-1P and ISBA-MEB in terms of surface temperatures, fluxes, etc. decrease as the cover decreases in height: the results converge as the surface becomes devoid of vegetation (as one would logically expect). So, results tend to be closer for grasses and annuals just like wheat than trees. This is because the main differences arise owing to the difference in the within-canopy turbulence treatment (both versions use the same functions based on MOST above the momentum sink point (z0 for ISBA-1P, z0+d for ISBA-MEB where d is the displacement height): as the d→0, the models converge to a certain extent. Also, we expect ISBA-MEB to perform best compared to ISBA-1P for tree canopies with moderate values of fraction cover just as the Olive orchard (moderately sparse). In contrast, as LAI becomes large, in ISBA-1P the vegetation weight Fc tends to 1, thus ISBA-1P resembles a completely vegetation surface and ISBA-MEB and ISBA-1P converge. This behavior can be seen, for example, in Napoly et al. (2017: Fig. 15b), ISBA-1P and ISBA-MEB differences (biggest improvements for MEB) are largest for LAI values in the range from 3 to 4 m2/m2, which corresponds to the LAI for the olive grove in this study. These results are based on an analysis of many FluxNet sites, and are consistent with several local sites in France (again, as detailed in Napoly et al., 2017). This arises mainly because the differences between the surface and the vegetation (temperatures, fluxes. . .) are most contrasted for sparse.

This means that, more specifically for our sites:

(1) For wheat, the added value of a double energy budget should be evident from emergence until full cover when vegetation is sparse. The surface can be considered homogeneous out of this period either considering bare soil at the start of the season or fully covering vegetation after. By contrast, from emergence to full cover, the cover sparsity may lead to a strong difference between soil and vegetation temperatures and some level of coupling between both energy sources but this period is short for wheat. It covers less than 1 month around march at +/- 10 days. In addition, irrigation tends

to limit such contrast between component temperatures thanks to uniform quite well-watered conditions. To our opinion, both irrigation and the shortness of the period when vegetation is sparse may explain the similar performances of the ISBA-1P model with regards to MEB.

(2) Over the Olive Orchard, ISBA-MEB performed well during the whole crop season because of the moderately open canopy of the field (fraction cover Fc is about 55%) meaning that both soil and vegetation sources are significantly coupled. The significantly worst performances (still considering LE) of ISBA-1P and to a lesser extent ISBA-2P are mainly attributed to the highly transient regime due to the flooding irrigation technic. Indeed, for ISBA-1P, there is no partition of net radiation between soil and vegetation, the high available energy (Figure 6) is used for soil evaporation at the time of irrigation (see Figure S4). This is a well known behavior of ISBA-1P in forest regions unless Fc→1. Stated differently, since with only one energy budget soil temperature is the same as the temperature of the vegetation, soil experience very little shading (in addition, it used the same -relatively large- z0 since the nonlinear aggregation of z0 for soil and vegetation tends to result in a z0 much closer to the higher elements -the vegetation z0-, as one would expect). The better behavior of ISBA-2P with regards to -1P is due to a compensation between both patches. Indeed, soil evaporation is significantly over-estimated at the time of the irrigation for the soil patch because of a direct exposure to incoming soil radiation and because there is no roots to extract water for transpiration on this patch. By contrast, soil evaporation is almost nil for the patch vegetation because of a complete screening by a dense vegetation cover (LAI=5 and Fc=1; Figure S4). Once the upper soil surface is dry (about two to three days after irrigation), ISBA-MEB, -1P and 2-P are close.

According to the reviewer's comment, this point is further argued all along the manuscript based on the explanations provided above (see colored version of the manuscript).

1.7. It is concluded that 1-P accuracy is "sufficient" for the wheat because both models

perform similarly, but a measurement of percentage error or average LE values is not provided. Without this information, it is not possible to get an idea of the real utility of these estimations.

Agree. According to the reviewer comment, the percentage errors are now provided in the new version of the manuscript. We decided to provide the values within the text to avoid burdening the table 4.

1.8. According to water inputs observed in 4, the wheat was barely irrigated during the first season. It should have suffered severe water stress, with a poor development. Could this issue have affected the calibration of the different models over wheat? It would be useful to interpret the results to add a brief description of crop conditions during the different seasons in the site description.

Agree. The water input applied in the 2003 season is very low compared to the amount provided to the field in 2013. Indeed, only four irrigation events were applied and were not well managed due to the technical constraint of the concrete channel network imposed by the institution in charge of agricultural water (ORMVAH; Haouz Agricultural Development Regional Office). However, the development of the wheat was almost normal. Indeed, Er-Raki et al. (2007) have found that the lengths of growing stages of this wheat compared well with those of an another field (six irrigation events) very near to our site. Over the same field, Boulet et al. (2007) have revealed that water stress occurred late in the season when senescence has already started (around May, 6th; the reason is that the farmer stopped the irrigation on April, 21st). However, this late stress cannot strongly affect the the calibration as it occurred during senescence. According to the reviewer suggestion, the crop conditions are now briefly described in the section describing the site.

1.9. Were olive groves maintained free of grasses all year round? The appearance of a grass layer between olive trees during part of the growing season is quite often. It could be an intended management practice or occur naturally and not be properly

removed. Either way, it would highly affect the balances of energy and water, and it should be mentioned

Agree. In general, the olive orchard was well managed in our site. In particular, the understory vegetation was removed regularly. We assumed that it has a low impact on the measurements. According to the reviewer comments, this is now explained in the description of the crop conditions. We have also added the tree spacing and the inter-row length.

1.10. A few sentences of the abstract are unclear (lines 31-40), with confusing and sometimes erroneous references to the different versions and crops. For example, in lines 31-33, it should be specified, within the sentence, that it makes reference only to the results on wheat, and 2P is not applied on wheat.

Agree. The abstract has been partly rewritten in response to the reviewer's comment as follows:

"The main objective of this work is to question the representation of the energy budget in surface-vegetation-atmosphere transfer (SVAT) models for the prediction of the turbulent fluxes in the case of irrigated crops with a complex structure (row) and under strong transient hydric regimes due to irrigation. To this objective, the Interaction Soil-Biosphere-Atmosphere (ISBA-A-gs) is evaluated over a complex open olive orchard and, for comparison purpose, on a winter wheat field taken as an example of homogeneous canopy. The initial version of ISBA-A-gs based on a composite energy budget (named hereafter ISBA-1P for 1 patch) is compared to the new multiple energy balance (MEB) version of ISBA representing a double source arising from the vegetation located above the soil layer. In addition, a patch representation corresponding to two-adjacent uncoupled source schemes (ISBA-2P for 2 patches) is also considered for the Olive orchard. Continuous observations of evapotranspiration (ET) with an eddy-covariance system and plant transpiration (Tr) with Sapflow and isotopic methods were used to evaluate the three representations. A preliminary sensitivity analyses showed

a strong sensitivity to the parameters related to turbulence in the canopy introduced in the new ISBA-MEB version. Over wheat, the ability of the single and dual-source configuration to reproduce the composite soil-vegetation heat fluxes was very similar: the RMSE differences between ISBA-1P, -2P and -MEB did not exceed 10 W/m2 for the latent heat flux. These results showed that a composite energy balance on homogeneous covers is sufficient to reproduce the total convective fluxes. The two configurations are also fairly close to the isotopic observations of transpiration in spite of a light underestimation (overestimation) of ISBA-1P (ISBA-MEB). On the Olive Orchard, contrasting results are obtained. The dual source configurations including both the uncoupled (ISBA-2P) and the coupled (ISBA-MEB) representations outperformed the single source version (ISBA-1P) with slightly better results for ISBA-MEB in predicting both total heat fluxes and evapotranspiration partition. Concerning plant transpiration in particular, the coupled approach ISBA-MEB provides better results than ISBA-1P and, to a lesser extent ISBA-2P with RMSEs of 1.60, 0.90, 0.70 mm/day and $R^2$ of 0.43, 0.69 and 0.70 for ISBA-1P, -2P and MEB respectively. In addition, it is shown that the acceptable predictions of composite convective fluxes by ISBA-2P for the Olive orchard are obtained for the wrong reasons as neither of the two patches is in agreement with the observations because of a bad spatial distribution of the roots and of a lack of incoming radiation screening for the bare soil patch. This work shows that composite convection fluxes predicted by the SURFEX platform as well as partition of evapotranspiration in a highly transient regime due to irrigation is improved for moderately open tree canopies by the new coupled dual-source ISBA-MEB model. It also points out the need for further local scale evaluation on different crops of various geometry (more open rainfed or denser intensive olive orchard) to provide adequate parameterization to global data base such as ECOCLIMAP-II in the view of a global application of the ISBA-MEB model."

In addition, references to the different version and crops have been checked all along the manuscript and corrected. In particular, the results of the ISBA-2P version on the wheat site have been discarded from table 5 to avoid confusion.

1.11. On the next sentence, starting "By contrast" it is not clear if the contrast is because of the crop or due to the model, as 2P is only applied to olive trees. On the next "By contrast" (line 37), it is not clear to which contrast the authors are referring.

Agree. See response to point 1.10.

1.12. Line 102. Kustas and Norman, 1997 or 1996? Please correct the reference if it is really useful. The paper presents a review of many models not specifying a patch representation. The other two references: Norman et al. 1995 and Boulet et al. 2015 make reference to both, parallel and series, schemes.

Agree. Thank you. There was a referencing error in the manuscript as we wanted to refer to Kustas and Norman (1997) while the paper from Kustas and Norman (1996) was in the list of references. In addition, we didn't find papers evaluating patch approaches only apart from several paper of the climate community discussions the representation of ecosystems within grid points of atmospheric models (Bonan et al.; 2002 for instance). We decided to remove Kustas and Norman (1997) that is redundant with Norman et al. (1995) and keep Boulet et al. (2015).

Kustas, P., Norman, J.M., 1997. A two-source approach for estimating turbulent fluxes using multiple angle thermal infrared observations. Water Resour. 33, 1495–1508.

1.13. Please check the wheat site coordinates. It probably should be 31âŮe38' instead of 31âŮe68'.

The reviewer is right. Thanks. The coordinates have been corrected.

1.14. Line 187: What "Similarly to R3 site" means here?

OK. The reviewer is right. It was probably a mismatch in previous copy/paste. Thank you.

1.15. Line 362. Did daily calculations include nighttime?

OK. Daily calculations refer to diurnal value from 9 to 17h. This is now detailed in the

new version of the manuscript (lines 399-400).

1.16. Conclusions. Line 567. -2P was not applied to wheat.

Agree. See response to point 1.10.

—————————————————————

---

## Author Comment (AC2) · 26 Mar 2020

We thank the reviewer for his in depth reading of the manuscript. All the typo and grammar errors reported in the annotated manuscript have been corrected. In addition, you will find below the answer to his comments.

1.1. "while water transfers are active on the 0-2m layer only" reformulate

[Figure]

OK. As the discretization of the soil column up to 12 m in ISBA is a trick to better predict the soil temperature profile, the second part of the sentence has simply been removed.

1.2. "Indeed, the appearance of a strong sensitivity of the two parameters and , seems to be consistent (Choudhury and Idso 1985) and optimal higher (lower) values than literature of ( ) are obtained (Table 3)." reformulate

OK. Done as follows:

"Indeed, the strong sensitivity of the two parameters and , seems to be consistent with Choudhury and Idso (1985). In addition, the optimal value of ( ) is higher (lower) than literature (table 3)."

1.3. "As a conclusion, the high sensitivity to the new parameters introduced in ISBA-MEB and the optimal values of the sensitive parameters being significantly different from default literature values, means that sstudies at the local scales should be duplicated to determine specific parameters values for for different eco- and agro-systems to feed the ECOCLIMAP-II data base with specific parameter values in the view of a large-scale applications." Unclear conclusion.

OK. Reworded as follows:

"As a conclusion, the optimal values of the sensitive parameters being significantly different from literature values, studies at the local scales should be duplicated to determine specific parameters values for for different eco- and agro-systems in the view of a large-scale applications."

1.4. "This is probably because the bare soil area between the tree rows (the inter-row is about 8 m.) is not sufficiently large to consider two independent heat sources located side by side." Clarify this sentence. OK. Reworded as follows:

"This is probably because the bare soil area between the tree rows (the inter-row is about 8 m.) is not sufficiently large to consider that soil and vegetation heat sources doesn't interact with each other by locating the two sources side by side."

1.5. "It is well known that part of the observed discrepancies between simulations and observations can be related to the eddy eddy-covariance measurements because of the associated strong heterogeneity within the footprint during an irrigation event." well the models are not super trustworthy neither - nuance that more.

Agree. Of course. The sentence has been moved at the end of the paragraph to complement the analysis of the model deficiencies around an irrigation events and it has been reworded as follows:

"In addition to the model deficiencies at the time of irrigation as already highlighted, part of the discrepancies between simulations and observations can be related to the eddy eddy-covariance measurements because of the associated strong heterogeneity within the footprint during an irrigation event."

1.6. "... dissipates much less energy by soil conduction compared to the other two configurations" Why ?

OK. The explanation was given in the next paragraph. This is now explained as follows:

"... dissipates much less energy by soil conduction compared to the other two configurations. This is due to a compensation between the soil and the vegetation patches as explained below."

1.7. "As for the latent heat flux, the dual sources configurations outperformed the single source version for sensible heat flux predictions over complex cover with wide differences in performances between ISBA-1P on one hand, and ISBA-2P and ISBA-MEB on the other hand over the olive orchard site (Figure 5)." Unclear sentence.

OK. Sentence reworded as follows:

"For the sensible heat fluxes, the dual sources configurations ISBA-2P and ISBA-MEB also outperformed the single source version ISBA-1P for sensible heat flux predictions over the olive orchard."

1.8. "… it appears that the patch bare soil dissipates a lot of energy by conduction as shown by the … " A lot compared to what ?

OK. Sentence reworded as follows:

"… it appears that the patch bare soil (like ISBA-1P) dissipates much more energy by conduction than ISBA-MEB as shown by the …"

1.9. "The most striking feature is the significantly higher energy available for convection and conduction at the soil level for ISBA-1P and ISBA-2P with regards to ISBA-MEB (the reverse is obviously true for vegetation net radiation, not shown)." clarify waht is meant by conduction and convdction – unclear

Agree. Sentence reworded to make it clearer as follows:

"The net radiation at the soil surface is obviously lower for ISBA-MEB than for ISBA-1P and -2P because of vegetation screening and real partition between the two sources."

1.10. "ISBA-1P lies in-between because the canopy is open (Fc=55%)." reformulate sentence

The sentence was discarded in the new version of the manuscript without changing the conclusion of the analysis.

1.11. "The transpiration measured by the Sapflow at the Olive orchard site was aggregated at a daily timescale and converted in mm/day." only sap velocity is measured - explain how you convert to transpiration

Agree. According to the reviewer's comment (and also in response to reviewer 1), the extrapolation from sapflow measurements to transpiration at the field scale was detailed at 2.1.4 Data description section "evapotranspiration partition".